# Modern-like deep water circulation in Indian Ocean caused by Central American Seaway closure

Priyesh Prabhat [1,2] ✉, Waliur Rahaman [1] ✉, Nambiyathodi Lathika [1], Mohd Tarique[1], Ravi Mishra[1] & Meloth Thamban[1]

Global overturning circulation underwent significant changes in the late Miocene, driven by tectonic forcing, and impacted the global climate. Prevailing hypotheses related to the late Miocene deep water circulation (DWC) changes driven by the closure of the Central American Seaways (CAS) and its widespread impact remains untested due to the paucity of suitable records away from the CAS region. Here, we test the hypothesis of the large-scale circulation changes by providing a high-resolution record of DWC since the late Miocene (11.3 to ~2 Ma) from the north-western Indian Ocean. Our investigation reveals a progressive shift from Pacific-dominated DWC before ~9.0 Ma to the onset of a modern-like DWC system in the Indian Ocean comprising of Antarctic bottom water and northern component water during the Miocene-Pliocene transition (~6 Ma) caused by progressive shoaling of the CAS and suggests its widespread impact.

Global Overturning Circulation (GOC) plays a critical role in controlling ocean heat distribution and atmospheric $CO_2$ levels and thereby influencing global climate[1–3]. Tectonically driven changes in the ocean-gateways since the late Miocene had a dramatic impact on GOC, such as the closure of the Central American Seaway (CAS) during the late Miocene[4,5] and the Indonesian Throughflow (ITF) during the past 3 to 4 Ma[6,7]. These tectonic changes might influence the formation of the northern component water [NCW, a precursor of North Atlantic Deep Water (NADW)] in the North Atlantic and Antarctic Bottom Water (AABW) in the Southern Ocean and their export to the global oceans. These large-scale changes in deep water circulation (DWC) might impact global climate through ocean-atmosphere $CO_2$ and heat exchange and provide feedback to the climate system. The prevailing hypothesis related to the ocean-climate coupling and their feedback through heat and $CO_2$ exchange since the late Miocene remains untested due to lack of suitable records of formations, export, and distributions of NCW and AABW in global oceans. To the best of our knowledge, these records are mostly from the Pacific and the Atlantic oceans; the majority of these records are either from proximity to the deep water formation regions or oceanic seaways. Hence these records

might not necessarily reflect the widespread impact and large-scale changes in deep water circulation. Due to the absence of any major deep water formations in the Indian Ocean, it acts only as a host for deep water circulation (NCW and AABW). Further, the northern Indian Ocean is located at one of the terminal ends of the GOC, far away from the deep water formation regions and oceanic seaways. These specific features of the northern Indian Ocean make it an ideal basin to assess large-scale deep-water circulation changes in the past associated with tectonics and/or climatic changes and to validate the hypotheses related to ocean-climate couplings and their feedback mechanisms since the late Miocene. Few studies have been carried out in the Indian Ocean to reconstruct past DWC on a longer timescale, based on authigenic neodymium isotope composition ($\varepsilon_{Nd}$) of sediment cores[8–10] and Fe-Mn crust records[11,12]. These Fe-Mn crusts, with the growth rate of 1 to 4.3 mm/Ma, are situated at deeper depths (>4000 m) bathed only by AABW and hence, suitable for the reconstruction of AABW only. On the other hand, the existing high-resolution authigenic $\varepsilon_{Nd}$ records are either from the intermediate depths of the Central Indian Ocean[8–10] or from the Bay of Bengal (BoB)[9,13] (Fig. 1, Supplementary Fig. 1d). The BoB records are known to

[1]National Centre for Polar and Ocean Research, Ministry of Earth Science, Goa, India. [2]School of Earth, Ocean and Atmospheric Sciences, Goa University, Goa, India. ✉e-mail: priyeshprabhat@gmail.com; waliur@ncpor.res.in

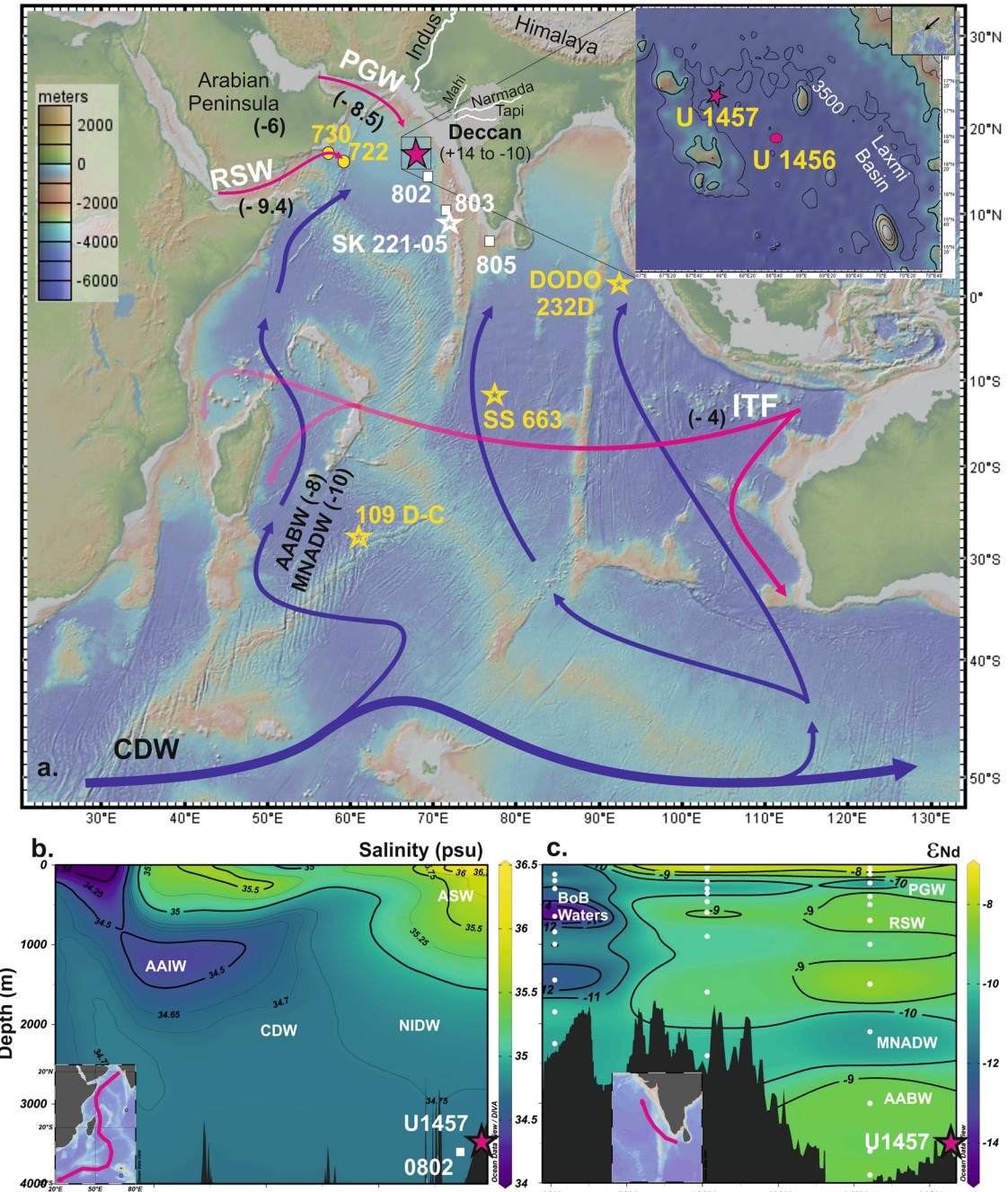

**Fig. 1 | Study Area and location of authigenic $\varepsilon_{Nd}$ records. a** Location map with the water mass pathways and their characteristics $\varepsilon_{Nd}$ values. The pink star represents core site (U1457) of the present study. The white square represents the proximal seawater stations 0802, 0803 and 0805 in the Arabian Sea[17]. The white star marks the location of sediment core site SK 221-05[16] used for reconstruction of authigenic $\varepsilon_{Nd}$ records in the Arabian Sea. Yellow star marks Fe-Mn crust sites DODO 232D[11], 109 D-C and SS 663[12] from the Indian Ocean. Yellow circle along the Oman coast marks the core site used for the paleo-monsoon reconstructions. Surface water masses are marked by pink lines and blue lines indicate deep water masses. The image in the inset shows the high-resolution bathymetry of the core site location. The map was produced using Geomap app (http://www.geomapapp.org); **b** A south-north transect (shown in inset) salinity section representing major water masses of the study area; **c** The $\varepsilon_{Nd}$ profile of the water stations present in the Arabian Sea along the south-north transect (shown in inset). The depth-salinity and $\varepsilon_{Nd}$ distribution maps were produced using Ocean Data View Software (https://odv.awi.de/). PGW- Persian Gulf Water, RSW- Red Sea Water, ITF- Indonesian Throughflow Water, AAIW- Antarctic intermediate water, NIDW- North Indian Deep Water, CDW- Circumpolar Deep Water, MNADW- Modified North Atlantic deep water, AABW- Antarctic Bottom Water, ASW- Arabian Sea Water (which includes PGW, RSW and Arabian Sea high salinity water), BoB waters- Bay of Bengal waters.

be affected by the release of particulate Nd supplied by the Himalayan rivers[14,15] and hence are not suitable for the reconstruction of DWC. Thus, it is imperative to have records from the regions not affected by such processes and represent large-scale circulation in the Indian Ocean. A recent study[16] has successfully reconstructed the DWC of the last glacial cycle (~136 ka) based on authigenic, and foraminifera $\varepsilon_{Nd}$

records from the eastern Arabian Sea (SK 221-05, 9°0'32" N, 72°5'32" E; 2700 m water depth) and thus suggests Arabian Sea would be a suitable location for the reconstruction of the past DWC.

In the present study, we have generated $\varepsilon_{Nd}$ record from the Arabian Sea at the core site U1457C (17°9.95' N, 67°55.81' E, 3522 m) and reconstructed the DWC record of the Indian Ocean for the interval

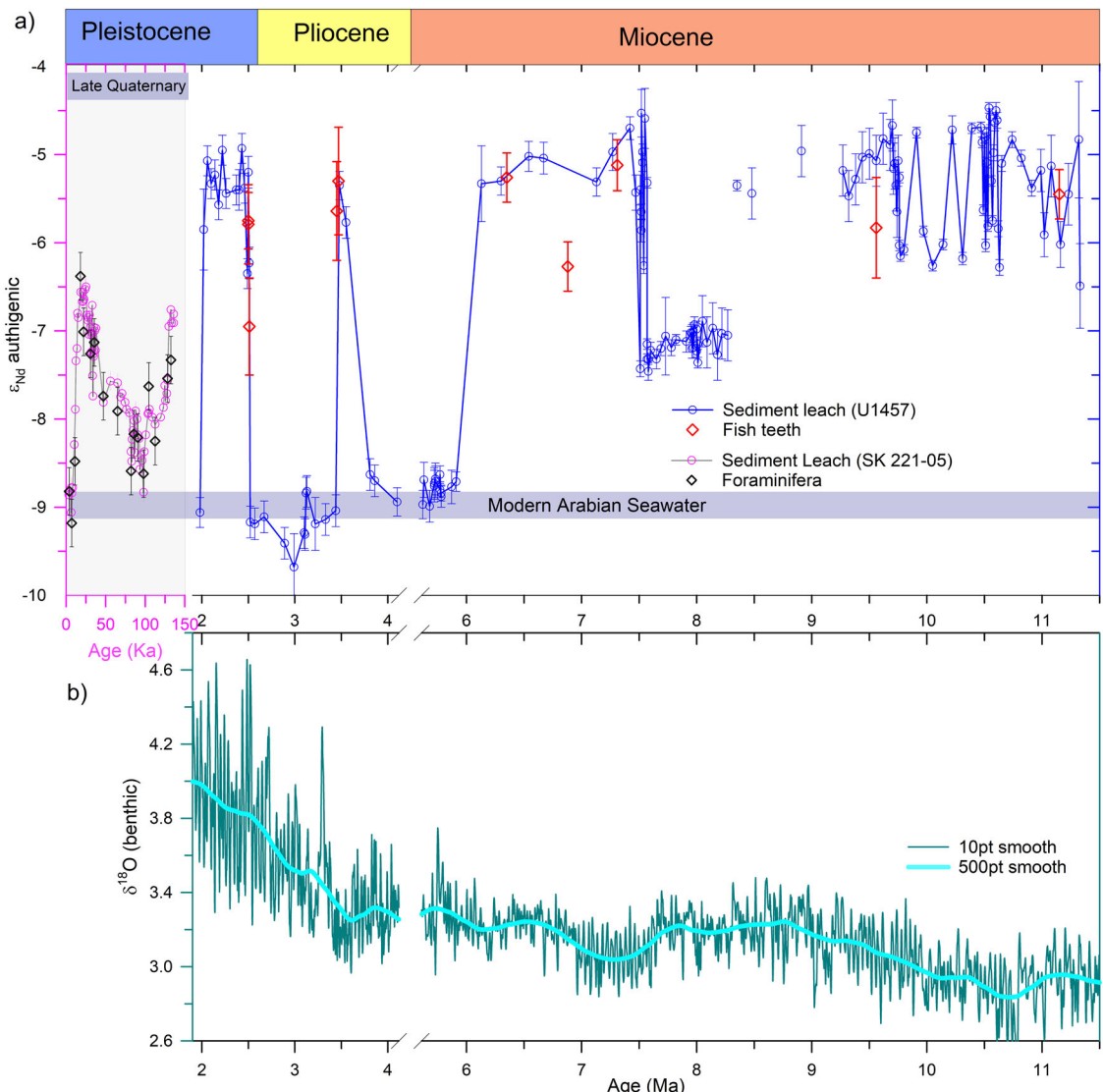

**Fig. 2 | Authigenic and detrital ε_Nd records from the Arabian Sea. a** Authigenic ε_Nd record of site U1457 (water depth 3522 m) covering a time span of 1.9–11.3 Ma (blue line), the late Quaternary record (pink line) is from site SK 221-05 (water depth 2700 m)[16]. The blue horizontal bar represents the modern-day deep water ε_Nd value for the Arabian Sea; **b** Global benthic δ¹⁸O record[40]. Error bar represents external error (2σ) of the ε_Nd measurements.

between the late Miocene to early Pleistocene (11.3–1.98 Ma). According to the modern hydrographic settings, the core site is presently bathed by Lower Circumpolar Deep Water (LCDW) with a higher proportion of AABW (Fig. 1)[17]. Thus, the ε_Nd record at this core site would be able to capture large-scale changes in the production and export of deep water masses in the late Miocene associated with tectonic and climatic changes. This will enable us to understand how the Indian Ocean DWC responded to these events in the past and evolved to the modern-like deep water circulation system.

## Results and discussion
### Authigenic and detrital Nd-isotope records
Nd isotope compositions were measured in the authigenic fraction of the bulk sediments (authigenic ε_Nd) and a few fish teeth/debris samples (*n* = 10) in a sediment core recovered from the IODP site U1457 C from the eastern Arabian Sea (Fig. 2a, Supplementary Table 2). In addition, Nd isotope composition was also measured in selected residual detrital fraction (detrital ε_Nd) of the same samples (Fig. 3f) to examine the diagenetic overprinting on the authigenic ε_Nd. The fish teeth ε_Nd is considered to be more robust and reliable[18] than leachate ε_Nd as a proxy for past water mass circulation due to the possibility of partial

dissolution of detrital Nd during the leaching of bulk sediments, which can alter the authigenic ε_Nd signature[19–21]. The agreement between the fish teeth and leachate ε_Nd values (Fig. 2a, Supplementary Fig. 3) gives us confidence that the leachate ε_Nd is not affected by the partial dissolution of detrital Nd during the leaching. Due to restricted occurrences of fish teeth and foraminifera in the present core, ε_Nd was measured in the leach fraction to generate a high-resolution record.

The authigenic ε_Nd record shows a narrow range from −6.5 to −4.5 with an average value of −5.5 ± 1 during the late Miocene interval (11.3 to 8.3 Ma). Thereafter, it shows a progressive shift towards less radiogenic values except for the three distinct excursions of more radiogenic values with similar magnitude (−5.5 ± 0.5) during the intervals of 7.4 to 6 Ma, ~3.5 Ma, and 2.5 to 2 Ma (Fig. 2a). The observed range in the authigenic ε_Nd record between −9 and −4.5 at the U1457 site is much higher than the earlier reports from the northeastern Arabian Sea and the equatorial Indian Ocean (EIO) (−9 to −6.5) during the late Quaternary glacial-interglacial periods[16,22]. The overall trend in the authigenic ε_Nd profile excluding these three excursions shows a progressive shift from more radiogenic values (−5.5 ± 1) prior to ~9 Ma to less radiogenic values (−8.7 ± 0.11) during the Miocene-Pliocene transition (~6 Ma) (Fig. 2a). These less radiogenic values are

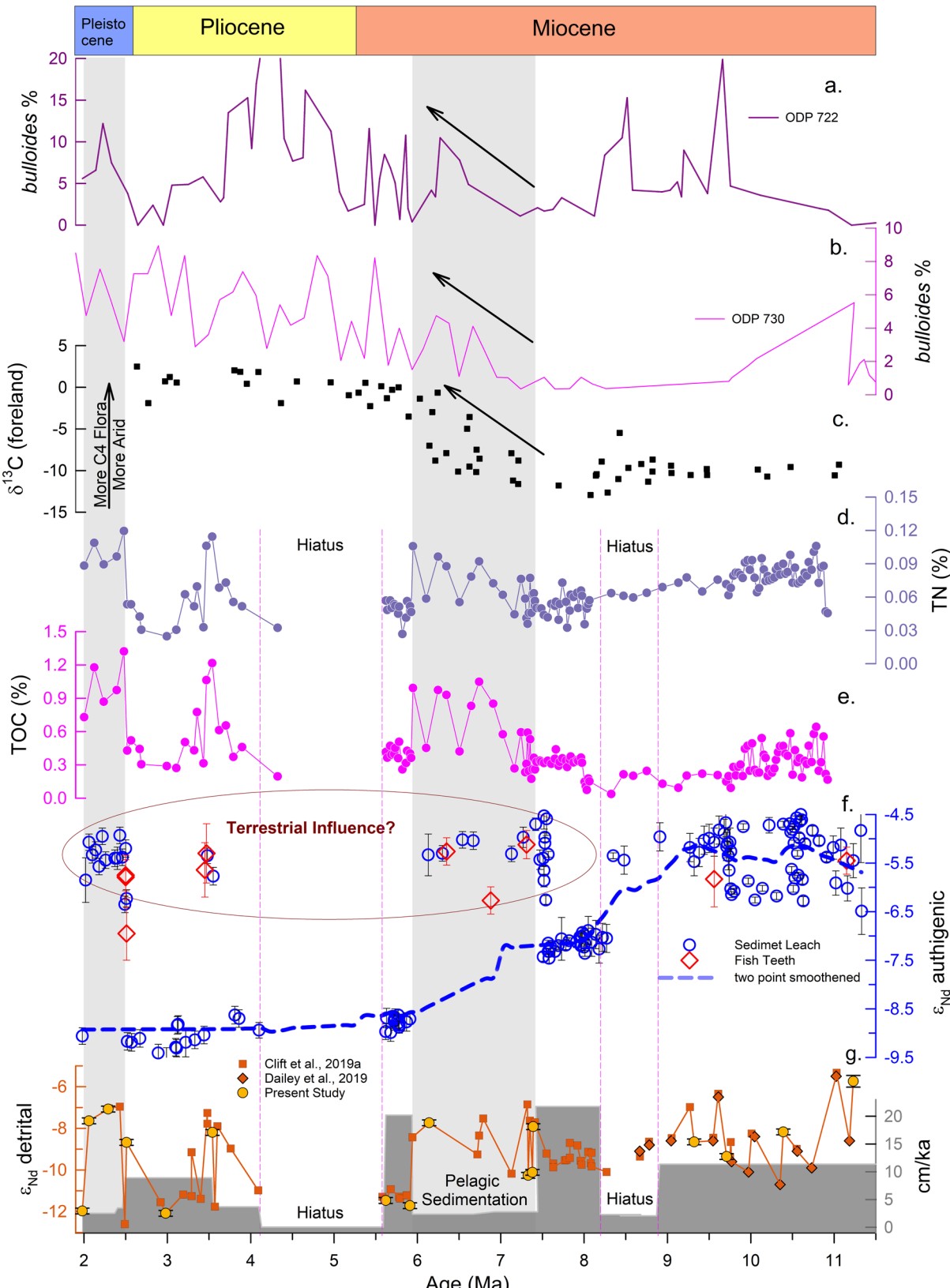

**Fig. 3 | Multi-proxy records and their comparison with the authigenic ε$_{Nd}$ record. a**, **b** *bulloides* % from the Oman Coast, Arabian Sea (recreated from ODP 722[32], ODP 730[33]); **c** δ[13]C from foreland Pakistan Siwalik[31], **d** Total Nitrogen (TN, weight %) record from Site U1457[27]; **e** Total organic carbon (TOC, weight %) record from Site U1457[27]; **f** Authigenic ε$_{Nd}$ record of site U1457 (present study), the dashed

blue line represents the two-point smoothened data to show the trend; **g** Sedimentation rate (cm/ka)[29, 64] and detrital ε$_{Nd}$ record (present study and published record[23, 24]). Error bar represents the external error (2σ) of the ε$_{Nd}$ measurements.

similar to that of the average value of the late Holocene −8.8 ± 0.2[16] and modern deep water −9 ± 0.4[17] reported from the Arabian Sea (Fig. 1, Supplementary Fig. 1c) and thus indicating that the authigenic $\varepsilon_{Nd}$ record has remained stable at the modern value (~−9) since the Miocene-Pliocene transition (~6 Ma). The detrital $\varepsilon_{Nd}$ measured in selected samples ($n = 16$), ranges from −12 to −5.7 (Fig. 3f, Supplementary Table 2), are consistent with the reported range (−12.6 to −5.3)[23,24] from this core (Fig. 3f).

We have conducted a detailed assessment to test the fidelity of the authigenic $\varepsilon_{Nd}$ record as a proxy for past deep water circulation in this oceanographic setting at the core site. This was conducted at two stages, i.e., (i) investigated the possible detrital dissolution during the leaching for the extraction of authigenic Nd from the bulk sediments and (ii) examined whether the authigenic Nd extracted from the bulk sediment recorded the deep water signature. A recent study from the South-eastern Arabian Sea[16] (Fig. 1a) has shown an excellent agreement between authigenic $\varepsilon_{Nd}$ of the leachate and foraminifera (Fig. 2a). The fish teeth/debris $\varepsilon_{Nd}$ data of the present core combined with the foraminifera $\varepsilon_{Nd}$ record of SK 221-05 from the South-eastern Arabian Sea[16] with their corresponding leachate $\varepsilon_{Nd}$ values show an excellent agreement over longer time scales ranging from the late Miocene to the Holocene (Fig. 2a, Supplementary Fig. 3). A similar study in the BoB[14] shows that authigenic $\varepsilon_{Nd}$ data extracted from bulk sediments using different leaching methods are similar to that of fish teeth and cleaned/uncleaned foraminifera within their analytical uncertainty. This close agreement between the $\varepsilon_{Nd}$ data of different archives suggests insignificant partial dissolution during the leaching process regardless of the methods used. Based on the present study and demonstration in earlier studies[14,16], we rule out the possibility of significant detrital dissolution during the leaching.

Modern water column Nd isotope study in the eastern Arabian Sea has demonstrated that deep water $\varepsilon_{Nd}$ signature primarily reflects the water mass mixing of AABW and NADW without significant contributions from other sources[17]. In the subsequent study, Lathika, et al.[16] successfully reconstructed the DWC history of the last glacial cycle (~136 ka) based on combined authigenic and foraminifera $\varepsilon_{Nd}$ records from the eastern Arabian Sea. This suggests that the extracted authigenic $\varepsilon_{Nd}$ at the U1457 core site records the past deep water $\varepsilon_{Nd}$ signature and hence can be used to reconstruct the past deep water circulation in the western Indian Ocean. However, contributions from other potential sources to the past deep water $\varepsilon_{Nd}$ variability must be investigated before interpreting this record in terms of deep water circulation.

## Factors contributed to the past authigenic $\varepsilon_{Nd}$ variability at the U1457 core site

A study[17] on dissolved Nd isotope in the eastern Arabian Sea water column has identified potential sources, their characteristic $\varepsilon_{Nd}$ signatures and quantified their contributions to the water column at various depths. The potential sources are (i) chemical weathering fluxes from nearby continents supplied by the westward-flowing rivers, (ii) aeolian dust deposition, and (iii) water mass mixing/circulations. Dissolved $\varepsilon_{Nd}$ data of the water column station 0802 (14.42 °N, 69.42 °E) (Fig. 1)[17] from the proximity to our core site have demonstrated that deep water $\varepsilon_{Nd}$ values at the similar core depth primarily reflect mixing of AABW (85%) and NADW (10–15%)[17]. Further, reconstruction of past deep water circulation from the eastern Arabian Sea[16] suggests that the composition of the Holocene and the inter-glacials water mass fractions were similar to that of modern-day, however, it increased to ~100% AABW during the glacials. It is noteworthy that the radiogenic values observed in our record ($\varepsilon_{Nd} = −4.5$) are even more radiogenic than the glacial values ($\varepsilon_{Nd} = −6.5$) when the northern Indian Ocean was completely occupied by AABW[16]. The radiogenic values (−6.5 to −4.5) in the older intervals (11.3 to ~8.3 Ma) do not show any significant correlation with the detrital $\varepsilon_{Nd}$ values. In contrast, there is a

good correspondence between authigenic and detrital values in the later intervals, particularly during the periods of radiogenic excursions Ca. 7.4 to 6 Ma, at ~3.5 Ma, and 2.5–2 Ma (Fig. 3f, g). The correspondence between the authigenic and detrital $\varepsilon_{Nd}$ values could be due to diagenetic overprinting of radiogenic material supplied by the Tapi River draining through Deccan basalts characterised by more radiogenic Nd[25], alteration of deep water through boundary exchange processes and/or partial dissolution of dust transported from northeast Africa and Arabian Peninsula. Since the core site U1457 is far from the continental margin and situated at a deeper depth of 3522 m, the possibility of boundary exchange processes for contributing radiogenic $\varepsilon_{Nd}$ values could be ruled out. Monsoonal precipitation influences sediment discharge and weathering fluxes from the Himalayan rivers, i.e., Indus and westward-flowing peninsular rivers such as Narmada, Tapi, and Mahi (Fig. 1a). The Nd isotopic composition of seawater might be influenced by the release of particulate Nd due to huge supply of sediments as it has been demonstrated in the BoB[14,15] and Andaman Sea[26]. However, a similar study in the Arabian Sea water column[17] and authigenic $\varepsilon_{Nd}$ record[16] from the Arabian Sea did not show any evidence for such release of particulate Nd altering deep water $\varepsilon_{Nd}$ signature. Though there are divergent views on the evolution of the south-west Indian monsoon, the most recent study based on multi-proxy records from the study area suggests a weak monsoon during our studied interval, which intensified after ~3 Ma[27,28]. It is noteworthy that more radiogenic values in the authigenic $\varepsilon_{Nd}$ profile coincide with the period of lower sedimentation rates[29] (Fig. 3g) and lower chemical weathering intensity[30] during the interval ~6 Ma to 7.7 Ma (Supplementary Fig. 4b), ruling out the possibility of enhanced weathering and erosion fluxes from the Deccan rivers. It is rather intriguing to note that these excursions of more radiogenic $\varepsilon_{Nd}$ values coincide with the higher percentage of total nitrogen (TN%) and total organic carbon (TOC%)[27], indicating higher productivity (Fig. 3d, e). These intervals of higher productivity further coincide with the increase in $C_4$ plant abundance based on $\delta^{13}C$ (foreland) record of organic carbon[31], indicating stronger aridity (Fig. 3c). Further, an increase in *G. bulloides* abundance (Fig. 3a, b) indicates enhanced upwelling driven by stronger wind conditions[32,33]. These compelling evidences based on multi-proxy records clearly suggest increased aridity over the potential dust source regions coupled with stronger wind-favoured enhanced dust supply to the Arabian Sea during these excursions. Therefore, the dust supply became the dominant fraction of the total sediment over the riverine supply during periods of lower sedimentation rates. The atmospheric dust deposition over the sea surface and release of Nd from aeolian dust can influence the isotopic composition of dissolved Nd[10]. The dust-derived $\varepsilon_{Nd}$ contributing to surface water at two sampling stations, 0802 and 0803 (Fig. 1a), were estimated to be −6.1 ± 2.2 and −5.3 ± 1.9, respectively[17]. These values are consistent with the reported values of major dust sources supplying to the Arabian Sea[34,35] (Supplementary Fig. 1b). Hence, we suggest that enhanced dust supply due to strong aridity could contribute to more radiogenic Nd and produce an agreement between detrital and authigenic $\varepsilon_{Nd}$ (Supplementary Fig. 5c). Therefore, we infer that these three radiogenic excursions resulted from the enhanced dust supply due to strong aridity over the continents and reduced river sediment supply that made the dust a dominant fraction of the terrigenous input. In conclusion, the radiogenic excursions resulted from enhanced dust supply and/or diagenetic overprinting and hence do not reflect water mass signature.

Excluding these extremely radiogenic values, the smoothed curve (blue dashed line, Fig. 3f) based on two points running average can be interpreted in terms of changes in the deep water circulations. This curve shows a systematic shift from more radiogenic $\varepsilon_{Nd}$ values during the late Miocene to less radiogenic values during the Miocene-Pliocene transition. This observation indicates a major change/reorganisation in the deep water circulation in the Indian Ocean, and hence we expect

similar observations in other $\varepsilon_{Nd}$ records elsewhere in the Indian Ocean. However, unfortunately, there is no $\varepsilon_{Nd}$ record available from the Indian Ocean at a similar depth range (Supplementary Fig. 1d). Alternate proxy records such as the benthic $\delta^{13}C$ records, have been extensively used to reconstruct the evolution of past water mass circulations despite several caveats attached to its application as a water mass circulation proxy[36,37]. The benthic $\delta^{13}C$ records from the Indian Ocean[32,33,38,39] shows a major negative shift between ~8 and 6.5 Ma, is consistent with the global benthic $\delta^{13}C$ records[40] (Supplementary Fig. 6), referred as "late Miocene carbon isotope shift" which has been attributed to the change in the terrestrial vegetation pattern from $C_3$ to $C_4$ plants and increase in the weathering inputs to global oceans[41]. Since the north Indian Ocean is among the most biologically productive basin and receives a huge amount of weathering fluxes from the Himalayan and Peninsular rivers, the benthic $\delta^{13}C$ would be largely affected by these processes and thus cannot be directly used as a proxy for water mass circulation. On the contrary, the $\delta^{18}O$ (benthic) record does not show a substantial shift in the values since the establishment of the West Antarctic ice sheet during the middle Miocene to late Pliocene[40] (Supplementary Fig. 7). The AABW formation enhanced after ~15.5 Ma due to the growth in the Antarctic ice sheet[42], and has been the densest water mass since the middle Miocene[43]. However, the benthic $\delta^{18}O$ record does not show concomitant changes with the global deep water circulation changes during the late Miocene. This indicates that the deep water mass circulation changes were decoupled from the Antarctic ice sheet evolution during the late Miocene.

The authigenic $\varepsilon_{Nd}$ record shows a discernible declining trend from more radiogenic values ($-5.5 \pm 1$) during the late Miocene (11.3 to ~9 Ma) to less radiogenic values ($-9 \pm 0.5$) during the Miocene-Pliocene transition, similar to that of the late Holocene and modern deep water in the Arabian Sea[16,17]. The radiogenic $\varepsilon_{Nd}$ values in the older intervals can be explained either by introducing additional deep water masses with more radiogenic $\varepsilon_{Nd}$ signature or by major changes/re-organisation of deep water circulation that modified the end-member $\varepsilon_{Nd}$ values of AABW and NADW. The Indian Ocean Fe-Mn crust records (109 D-C, water depth 5700 m, SS-663, water depth 5300 m, and DODO-232D, water depth 4119 m)[11,12] bathed by AABW in the modern time show uniform $\varepsilon_{Nd}$ values within a narrow range ($-8 \pm 1$) during the interval between ~2 to 16 Ma (Supplementary Fig. 2). This implies that the $\varepsilon_{Nd}$ values of AABW in the Indian Ocean were stable at around the modern-day value ($-8 \pm 1$) for the entire interval. Further, the authigenic $\varepsilon_{Nd}$ records from the Angola basin, the Fe-Mn Crust DS43[44] record (water depth 1990 m, bathed by upper NADW), and the sediment core ODP 1262[45] (water depth 4755 m, bathed by lower NADW), oscillated between $-13$ and $-11$ during our studied interval (Supplementary Fig. 8). These values are consistent with the modern NADW $\varepsilon_{Nd}$ range reported from the Angola basin (Supplementary Fig. 8). This also indicates that the $\varepsilon_{Nd}$ value of the NCW was within the range of modern-day $\varepsilon_{Nd}$ value of the NADW. The radiogenic values observed in the U1457 record ($-4.5$ to $-6.5$) during the older intervals (~11.3 to ~9 Ma) were higher than the modern deep water values and even higher than the Quaternary glacial-interglacial ranges[16]. These radiogenic $\varepsilon_{Nd}$ values in the Indian Ocean cannot be explained in terms of mixing of these two water masses alone (AABW and NCW, $\varepsilon_{Nd}$ end members stable and less radiogenic) and therefore hints at the possibility of major reorganisation in the deep water circulation in the southern ocean that modified to more radiogenic deep water $\varepsilon_{Nd}$ signature exported to the Indian Ocean.

The published records from the Arabian Sea[16], equatorial Indian Ocean[22], and South Atlantic[46] have revealed that the glacial values of CDW were modified to more radiogenic values ($-6.5 \pm 0.5$) due to reduced export of NADW to the Southern Ocean and enhanced supply of the Pacific Deep Water (PDW)[16,22] (Fig. 2a). The present tectonic configuration of the Arabian Sea has been stable since the closure of the Tethys Sea at ~14 Ma ago[47]. Therefore, the possibility of any

tectonically driven regional circulation changes within the Indian Ocean during our study interval (11.3 to ~2 Ma) could be ruled out. Among all the major water masses in the global ocean, Pacific water is characterised by the most radiogenic $\varepsilon_{Nd}$ values ($-3.5 \pm 0.5$)[2,48]. Therefore, the only possibility that can explain such radiogenic values of the deep Indian Ocean water is, the export of the Pacific water into the Indian Ocean. In the modern oceanographic setting, the Pacific water is exported to the Indian Ocean via two pathways, the Indonesian Throughflow (ITF) as surface water mass and the Southern Ocean as deep water mass (PDW). The Pacific water entering through the ITF is fresh and shallow water mass and hence cannot penetrate deeper depths up to ~ 3500 m to alter the deep water $\varepsilon_{Nd}$ signature at the present core site. Thus, we rule out the possibility of the Pacific water influence through the ITF to alter the deep water $\varepsilon_{Nd}$. Other possibility is the export of PDW to the Southern Ocean and mixing with CDW as recirculated Pacific Deep Water (rPDW). Previous studies of the last glacial cycle suggest that the rPDW occupied the deeper depth during the glacial period due to the shoaling/reduced export of NCW to the Southern Ocean[16,22]. Therefore, the enhanced contribution of rPDW to the Southern Ocean could make the CDW more radiogenic, and the export of this water mass to the northern Indian Ocean could produce such radiogenic $\varepsilon_{Nd}$ values.

## Forcing factors and mechanisms for the late Miocene circulation changes

Several studies based on models[49] and geochemical proxies[37,50] have reported weak NCW with reduced southward export during the late Miocene (Fig. 4d). The NCW export to the Southern Ocean was reduced due to weaker Atlantic Meridional Overturning Circulation (AMOC) caused by a higher influx of fresh and warm Pacific water into the mid-latitude Atlantic Ocean through the Central American Seaway (CAS)[51,52] (Fig. 5). Therefore, given the timing of the changes observed in the authigenic $\varepsilon_{Nd}$ record, constriction and shoaling of the CAS could be the most likely cause for the reduced influx of the Pacific water into the Atlantic Ocean and strengthening of NCW production after 9.5 Ma[37,53,54]. The less radiogenic shift of $\varepsilon_{Nd}$ records from the Caribbean Sea has been interpreted as the reduced influx and advection of Pacific water into the north Atlantic as a result of progressive closure and shoaling of the CAS after 9.5 Ma[53,54] (Fig. 5a). This conclusion was further supported by the model results[49]. Our observations of deep water circulation changes in the western Indian Ocean are consistent with the tectonically driven changes in the Pacific-Atlantic gateways, i.e., closure of the CAS (Fig. 4a). Therefore, the more radiogenic values oscillating between $-4.5$ to $-6.5$ during the interval of 11.3 to ~9 Ma could result from reduced export of NCW, compensated by the enhanced contribution of rPDW into the CDW. These evidences indicate that the CDW exported to the Arabian Sea comprised of a higher percentage of rPDW with more radiogenic $\varepsilon_{Nd}$. Subsequently, the authigenic $\varepsilon_{Nd}$ values gradually became less radiogenic during the interval of ~9 to 6 Ma with the progressive closure of CAS and thereafter became stable at $-9 \pm 0.2$, similar to the late Holocene ($-8.8 \pm 0.2$)[16] and modern deep water values ($-9 \pm 0.4$)[17] (Fig. 2a, Supplementary Fig. 1c). Therefore, we suggest that the progressive closure of CAS and thereby strengthening of AMOC after ~9 Ma enhanced the export of NCW to the Indian Ocean and attained modern deep water $\varepsilon_{Nd}$ value during the Miocene-Pliocene transition. The late Miocene cooling[55] of sea surface temperature at higher latitudes (Fig. 4b) might contribute to the increase in NCW formation and its density through brine rejection. Thus, the increased salinity along with the decrease in the bottom water temperature[56,57] (Fig. 4c), indicates the evolution of the density driven water mass mixing proportion of the deep water circulation in the study area and corroborates our conclusion. Further, the suggestions of enhanced production and export of the NCW to the Southern Ocean since the late Miocene based on benthic $\delta^{13}C$ records[37,49,50] and stable NCW formation during the

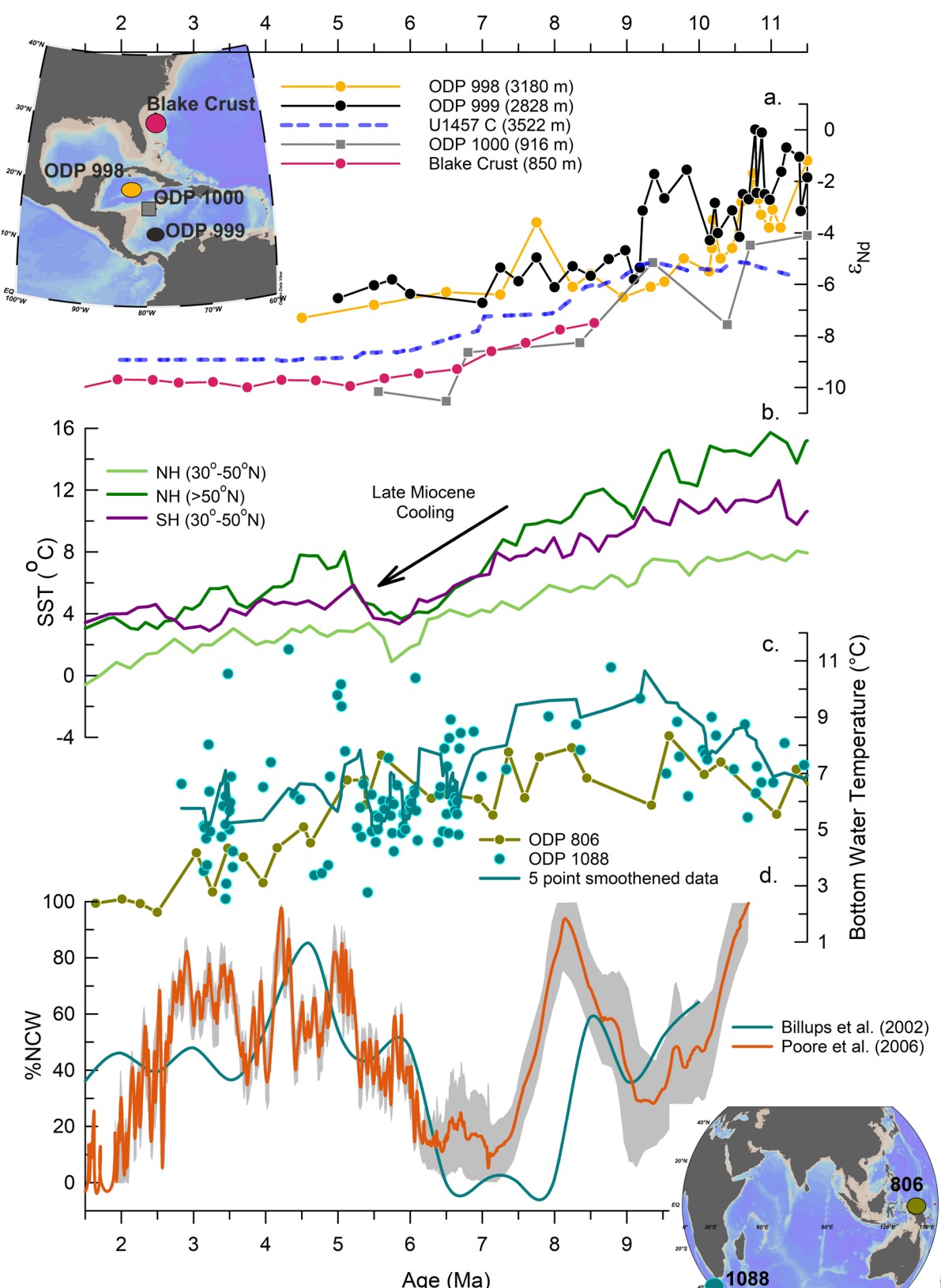

**Fig. 4 | Evolution of deep water circulation since the late Miocene. a** $\varepsilon_{Nd}$ record from the Caribbean Sea, Atlantic Ocean ODP 998, 999[54], ODP 1000[53], Blake Fe-Mn Crust[44]; **b** Sea surface temperature (SST) for the subtropical and high latitude northern hemisphere (NH) and southern hemisphere (SH)[55]; **c** bottom water temperature based on Mg/Ca paleothermometry[56,57]; **d** %NCW[37,50] estimation based on benthic $\delta^{13}C$. Grey envelope represents the error band on the %NCW record (Poore, et al.[50]).

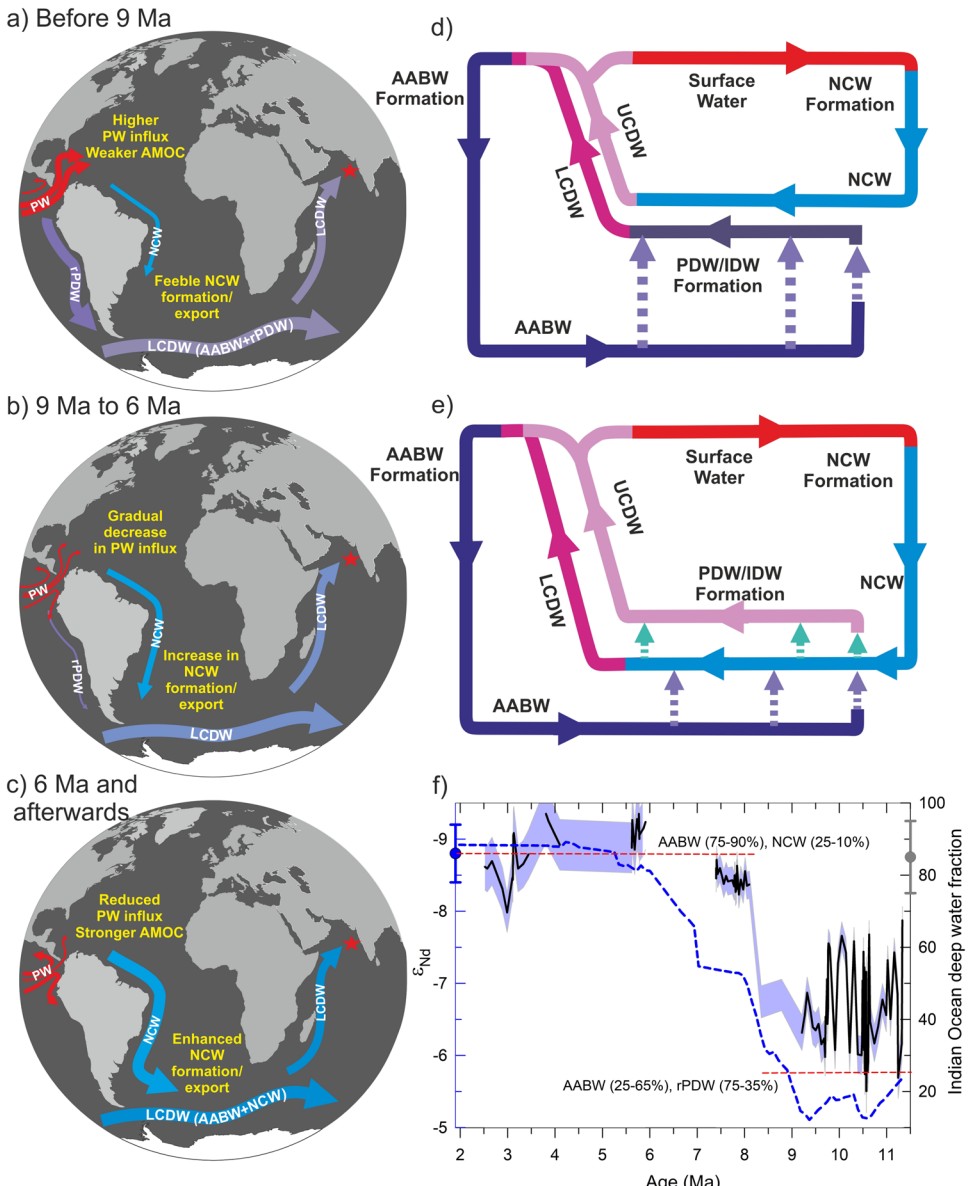

**Fig. 5 | Constriction and shoaling of the CAS and its widespread impact on deep water circulation in three different stages. a** Stage 1 (before ~9 Ma): large influx of fresh PW into the north Atlantic Ocean resulted in weaker AMOC and reduced formation and export of NCW to the Southern Ocean, as a result AABW mixed with rPDW to form the LCDW; **b** Stage 2 (-9 Ma to 6 Ma): gradual decrease in the PW influx to the North Atlantic and increase in the NCW formation and export to the Southern Ocean; **c** Stage 3 (after ~6 Ma): reduced export of PW into the north Atlantic and subsequent strengthening of NCW formation and enhanced export (as a result of stronger AMOC) to the Southern Ocean resulted in LCDW being formed by the mixture of AABW and NCW, while the rPDW formed the upper cell of the CDW owing to its lesser density than NCW similar to modern time; these schematic shows circulation cell structure (**d**) during the reduced NCW formation and its export to the Southern Ocean; **e** enhanced NCW formation and export to the Southern Ocean; **f** authigenic $\varepsilon_{Nd}$ record (two-point smoothened, blue line) of site U1457 and its corresponding Indian Ocean deep water fractions (black line with grey error envelope) with their modern-day values plotted on their respective axis. Horizontal red dotted lines show the contribution range of water masses at time interval before ~9 Ma (AABW and rPDW) and after ~9 Ma (AABW and NCW). PW - Pacific water; NCW - Northern component water; AABW - Antarctic bottom water; rPDW - recirculated Pacific deep water; PDW - Pacific deep water; IDW - Indian deep water; CDW - Circumpolar deep water; LCDW - Lower circumpolar deep water; UCDW - Upper circumpolar deep water.

final phase of the closure of CAS[58] (4.7 Ma and afterward) are consistent with the finding in the present study.

Due to the cessation of significant deep water exchange with the Pacific and the Atlantic Ocean through the CAS, the South Atlantic seafloor started bathing in NADW, which caused the deepening of the lysocline, carbonate compensation depth (CCD) and enhanced the carbonate preservation[59–61]. A recent compilation of calcium carbonate ($CaCO_3$) records from multiple DSDP and ODP sites in the Atlantic basin has demonstrated increased calcium carbonate sedimentation rates since the late Miocene throughout Atlantic[62] (Supplementary

Fig. 9c). One of the sites (ODP 1262) in the South Atlantic at the Walvis Ridge shows an increase in the $CaCO_3$ from ~10% since ~10 Ma to ~90% at ~7 Ma during the late Miocene and became stable thereafter (Supplementary Fig. 9b.) and suggested enhanced NCW in the South Atlantic[61]. This change in $CaCO_3$ (weight %) at site 1262 is coeval with the shift in $\varepsilon_{Nd}$ values toward less radiogenic signature[45] reported from the same site (Supplementary Fig. 8). This indicates that carbonate preservation depth started to increase since the past ~10 Ma, which became stable in the late Miocene (~7 Ma) under certain physico-chemical conditions. This evidence suggests that modern-day like

vertical structure/configuration of deep water masses in the Atlantic was achieved during the late Miocene. The $\varepsilon_{Nd}$ based reconstruction of DWC is consistent with the changes in lysocline and CCD in the Atlantic and thus corroborates our conclusion.

The deep water circulation changes driven by the tectonic forcing that contributed to the gradual shift from more radiogenic to less radiogenic $\varepsilon_{Nd}$ values in the Indian Ocean are illustrated in three stages (Fig. 5). The first stage before ~9 Ma (Fig. 5a) depicts a wider opening in the CAS, indicating more influx of Pacific water to the North Atlantic that resulted in weaker AMOC and reduced NCW export to the Southern Ocean. In the absence of NCW at deeper depths in the Southern Ocean, rPDW mixed with AABW and exported to the north-western Indian Ocean from the lower cell of the CDW (Fig. 5d). This circulation pattern changed with the progressive constriction of the CAS after ~9 Ma (stage 2), as depicted in Fig. 5b, where a gradual reduction in Pacific water influx into the North Atlantic resulted in enhanced formation of NCW and its southward export (Fig. 5b). During the later phase (~6 Ma, stage 3), CAS shoaled to a critical depth leading to reduced advection of the Pacific water in the Atlantic. This led to the strengthening of AMOC, which resulted in the enhanced formation and southward export of the NCW (Fig. 5c). Enhanced export of NCW to the Southern Ocean replaced rPDW from the lower cell of the CDW as rPDW shoaled due to its lower density compared to NCW and joined the upper cell of the CDW (Fig. 5e). Thus, after ~9 Ma, there was a gradual reduction in the rPDW component to LCDW, while AABW and NCW became the main component of the LCDW. This transition from the Pacific-dominated deep water circulation system to the AABW-NCW dominated circulation system in the Indian Ocean was gradual and finally stabilized at a modern-like circulation system during the Miocene-Pliocene transition at ~6 Ma.

We have quantified the proportion of each water mass (AABW, NCW, and rPDW) using the binary mixing model for different time intervals (see Method section). It is important to mention that in the mixing calculation, fraction estimates are critically dependent on how accurately the end member values were constrained. The end-member compositions might undergo significant changes over the studied interval, therefore, we have considered dynamic end-members for specific intervals with an uncertainty that will accommodate the variability and changes in the end-member values. The details of assigning end-member values and water mass mixing calculation are discussed in the method section and supplementary information. Authigenic $\varepsilon_{Nd}$ record shows a similar value to that of the modern and interglacial periods since ~6 Ma, and thus our semi-quantitative estimate shows deep water composition was stable at modern-like deep water composition comprises of 85–90% AABW and 10–15% NCW, since the Miocene-Pliocene transition. On the contrary, the DWC was dominated by the southern sourced AABW with a significant fraction of PDW (50 ± 10%) during the older intervals (11.3 to ~9 Ma) (Fig. 5f).

The late Miocene evolution trend of the deep water circulation shows a clear shift from the Pacific-dominated deep circulation system to the Atlantic influenced deep water circulation in the Indian Ocean with the strengthening of AMOC during the Miocene-Pliocene transition. Our finding suggests a profound and widespread impact of the late Miocene CAS closure on the evolution of ocean deep water circulation and validates the so called "Panama Closure Hypothesis".

## Methods
### Regional tectonic settings
Details of the regional tectonic settings of the study area have already been discussed elsewhere[29,63]. Briefly, the core site U1457 C lies on the western end of NW-SE oriented Laxmi basin in the eastern Arabian Sea[29]. The Laxmi Basin is a rift basin situated between the Laxmi Ridge in the west and the passive margin of the west coast of India[63] occupied by the distal fan of the Indus delta. The sediments of this region are mostly supplied by the Indus River and a few

westward-flowing peninsular rivers. The basement of the Laxmi basin is formed by basalt, while the oldest recorded sediments date to the late Palaeocene, while the youngest sediments represent the Holocene. The entire deposition history at this core site reveals major breaks in sedimentations (hiatus)[29,64]. Within our study interval, two hiatuses have been recorded, during the late Miocene (~8.7 Ma) and at the Miocene-Pliocene boundary (~5.5 Ma) (Fig. 3g, Supplementary Fig. 11).

### Hydrographic settings
Water mass structure and the oceanographic conditions of the Arabian Sea have already been discussed extensively in previous studies[65–67]. Briefly, the upper water column in the Arabian Sea up to 1000 m is occupied by the Arabian Sea High Salinity Water (ASHSW, potential temperature 24 °C to 28 °C, and salinity 35.3 to 36.7), the Persian Gulf Water (PGW, potential temperature 18 °C, and salinity 36.8), the Red Sea Water (RSW, potential temperature 11 °C, and salinity 36)[66,67]. The ASHSW is formed by excessive evaporation over precipitation of Arabian Sea water and occupies the top 50–100 m of the water column. Whereas the PGW and RSW are formed due to spillage of high salinity waters from the Persian Gulf and the Red Sea to the Arabian Sea, respectively. The intermediate water column is occupied by the mixture of ASHSW, PGW, and RSW during their southward flow and is known as the North Indian Intermediate water (NIIW), occupying a depth up to 1500 m. During its southward journey, it mixes with the upwelling polar waters to form the North Indian Deep Water (NIDW). At deeper depths, the Arabian Sea is ventilated from the south by the Modified North Atlantic Deep Water (MNADW, potential temperature 1.8 °C–2.8 °C and salinity 34.78–34.85) and Antarctic Bottom Water (AABW, potential temperature 0.3 °C and salinity 34.5)[65]. The MNADW and AABW enter the Madagascar and Mascarene basin as deep western boundary current (DWBC) through the Crozet Basin (30°S and 60°E). After reaching the Somali basin through the narrow Amirante passage, MNADW and AABW enter the Arabian Sea via Owen fracture Zone[65].

### Sediment Core
Sediment core was recovered from the site U1457 C (17°9.95' N, 67°55.81' E, 3522 m) during the IODP 355 expedition to the eastern Arabian Sea on the western edge of the Laxmi Basin (Fig. 1a). The core site was drilled on the Indus Delta fan and recovered ~917 m long core with sediment recovery of 48%[29]. The Indus river is the major contributor of sediments to the sampling site, along with the westward-flowing Narmada and Tapi rivers[29]. Lithology at the site is affected by the turbidite sequence dominated by dark grey to greenish-grey claystone and light brown to dark grey sand/sandstones[29]. The lower section below 830 m has the occurrence of calcarenite, breccia, and limestone. Occurrences of nannofossil chalk and nannofossil-rich claystone alternate with silty claystone and silty sandstone[29].

### Age models
The age model for site U1457 C is based on the published biostratigraphy[29,64] (Supplementary Table 1 and Supplementary Fig. 11).

### Nd isotopes as a proxy for deep water circulation
Nd has a residence time of 300–1000 years[68] in the ocean which is lower than the global ocean mixing time (~1500 years)[69] and behaves quasi-conservatively in the seawater column. Therefore, the authigenic Nd isotope composition of the sediment can be used for semi-quantitative estimates of past water mass mixing[70]. The Nd isotope ($\varepsilon_{Nd}$) is a reliable proxy for the reconstruction of water mass mixing, as it is not affected by biological and/or low-temperature processes[8,22,70]. The Nd isotopic composition is expressed as $\varepsilon_{Nd} = [(^{143}Nd/^{144}Nd)_{sample}/(^{143}Nd/^{144}Nd)_{CHUR} - 1] \times 10^4$, where CHUR (Chondritic Uniform Reservoir) has $^{143}Nd/^{144}Nd$ ratio 0.512638[71].

## Sample Preparation

Nd isotopes were measured in the authigenic and detrital phases of the bulk sediments as well as in few fish teeth /debris samples. Neodymium associated with the authigenic phase of the bulk sediment was extracted through acid reductive leaching following the method of Wilson, et al.[72]. Briefly, about 3 g of powdered sediment was leached with 10 ml of 0.02 M Hydroxylamine Hydrochloride (HH) in 4.4 M acetic acid (pH adjusted to 2) for one hour in a 15 ml centrifuge tube. The sample was then centrifuged at 5000 rpm, and the leachate was decanted carefully without the transfer of any residue. The same method was employed in a recent study from the South-eastern Arabian Sea[16] with similar regional settings to understand the deep water circulation of the last glacial cycle, which has demonstrated the reliability of the leaching method in extracting the seawater Nd from the bulk sediment. The Nd measured in fish teeth, and foraminifer is considered more robust and reliable than the leachate phase to reconstruct deep water circulation. Due to the lack of foraminifera in the studied core, we have measured Nd in a few handpicked fish teeth/ debris samples ($n = 10$) following Basak, et al.[18]. Initially, the fish teeth/ debris was cleaned repeatedly by ultrasonication in optima grade methanol. The cleaned samples were then treated with a 1:1 solution of $H_2O_2$ (30%) and Milli-Q, followed by dissolution in a 1:1 mixture of $HNO_3$ and HCl.

To extract the detrital fraction, the residual material after removing the authigenic phase was further treated with 0.6 N HCl for the complete removal of carbonates. Subsequently, samples were washed thoroughly with Milli-Q water and dried in the oven at 80 °C. Dried samples were ashed at 600 °C in a muffle furnace to remove organic matter. ~100 mg powdered, ashed samples were digested in precleaned Teflon vials using HF-$HNO_3$-HCL mixture at 120 °C following the method described in Tripathy, et al.[73]. Standard Reference Materials BCR-2 and BHVO-2 were also digested along with the samples.

Dissolved samples were purified by column chromatography and were passed through columns filled with cation-exchange resin AG50W-X8 (200–400 micron) to separate Rare Earth Elements (REE). REE fractions were then passed through the column containing Eichrom LNspec™ (50–100 micron) resin to separate Nd from the REEs[74].

## Nd isotope analysis

Nd isotope ratios were measured using Multi-collector Inductively Coupled Plasma Mass Spectrometer (Thermo Fischer Scientific, Neptune Plus) at National Centre for Polar and Ocean Research, Goa. The measured $^{143}Nd/^{144}Nd$ ratios were mass-bias corrected using $^{146}Nd/^{144}Nd$ ratio of 0.7219 and normalised to the reported JNdi-1 standard $^{143}Nd/^{144}$ value of 0.512115[75]. To ensure the quality of measurement, the international standard JNdi-1 was measured at every five samples, and the obtained average ratios were 0.512101 ± 8 ppm (2σ, $n = 42$) for the leachate, 0.512058 ± 14 (2σ, $n = 5$) for the fish teeth and 0.512125 ± 9 ppm (2σ, $n = 6$) for the detrital samples. The external reproducibility calculated for each session for the leachate, fish teeth, and detrital samples were 0.06–0.29 $\varepsilon_{Nd}$, 0.28 $\varepsilon_{Nd}$, and 0.14–0.28 $\varepsilon_{Nd}$ (2σ) units, respectively. If the internal error (2σ) is larger than the external error, the internal error is reported as the final uncertainty associated with the individual measurements. Several procedural blanks were also processed along with the samples and ascertained an average blank of ~110 pg for the leachate ($n = 10$), 13 pg for the fish teeth ($n = 1$), and 50 pg for the detrital phase ($n = 2$), which are three orders of magnitude lower than the total Nd typically analysed in samples. Hence, no blank correction was applied. Replicates have an average $\varepsilon_{Nd}$ variation of ±0.16 (2σ, $n = 15$) (Supplementary Fig. 10).

## End-member calculation

To quantify the fractional contributions of each water mass during the studied interval in the Arabian Sea, we have done Nd isotope mass balance calculation following the method of Rahaman, et al.[76]. The calculation is based on the assumption that the Nd isotope behaves quasi-conservatively. We have divided our record into two stages, and the water mass fractions of each stage are calculated separately. In the first stage (11.3 to ~9 Ma), the end member water masses taken are the AABW and PDW. In the second stage (~8 Ma onwards), mixing calculation is done considering the AABW and the NADW as the end members. The end-member composition of water masses for each stage is calculated using the following binary mixing equation. The uncertainty associated with the estimates was calculated using the Monte-Carlo error propagation method[76].

$$\varepsilon_{Nd_M} = \frac{\varepsilon_{Nd_A} * C_A * f_A + \varepsilon_{Nd_B} * C_B * f_B}{C_A * f_A + C_B * f_B} \qquad (1)$$

$$f_A + f_B = 1 \qquad (2)$$

The %A $\varepsilon_{Nd} = f_A*100$ is the relative contribution of the water mass A to the Arabian Sea, and %B $\varepsilon_{Nd} = 100 - $ %A $\varepsilon_{Nd}$ is the relative contribution of water mass B to the Arabian Sea. Here $C_A$ and $C_B$ represent the concentration of Nd in the water mass A and B, respectively. $\varepsilon_{Nd\,M}$ is the value of Nd isotope compositions of sediment leach from the Arabian Sea, and $\varepsilon_{NdA}$ and $\varepsilon_{NdB}$ are the end-members of the water mass A and B, respectively. The $f_A$ and $f_B$ represent the fractions of the water mass A and B. The details of end-member water masses and their end-member values used in both stages are provided in the supplementary (Table S3).

## Data availability

All the data generated for this study are provided in the supplementary file.

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

## Acknowledgements

Authors acknowledge the former Director Dr. M. Ravichandran, National Centre for Polar and Ocean Research, Ministry of Earth Science, Govt. of India for financial support through the "PACER- Cryosphere and Climate" project. We sincerely acknowledge shipboard scientific and technical teams of IODP expedition 355 and IODP-India. The authors thank Ankita Kanetkar for the laboratory support. P.P. and M.T. acknowledges the University Grant Commission for providing the research fellowship grant. This is NCPOR contribution number: J-49/2022-23.

## Author contributions

P.P. and W.R. designed the study. P.P., L.N. and W.R. wrote most of the text. P.P. and M.T. analysed the Nd isotopic composition. R.M. participated in the expedition and collected the samples. P.P., W.R., L.N., M.T., R.M. and M.Th. contributed to the result interpretation, discussion, and improvement of the paper.

## Competing interests

The authors declare no competing interests.
