## [Peer Review File · Nature Communications]

Onset of modern-like deep water circulation in Indian Ocean caused by Central American Seaway closureReviewer #1 (Remarks to the Author):

Noteworthy results: The manuscript titled "Onset of modern-like deep water circulation in the Indian Ocean caused by Central American Seaway closure" by Prabhat et al. contains a lot of valuable data and provides new insights about the onset of deep ocean circulation using the neodymium radiogenic isotopes. It's an impressive feat with the vast set of data authors provide to illuminate the resumption of the Miocene circulation on a tectonic scale. The authors accurately claim that there are NO high-resolution ϵNd data in the Indian Ocean for the Miocene period compared to equivalent data in the Atlantic Ocean at three ODP Caribbean and one Blake Crust site. The authors placed their data into a broader context using published data mainly from the Atlantic Ocean to very convincingly argue for the 11.5 and 7.5 Ma period. One of the other interesting findings is the ϵNd values between 6 and 3.8 Ma and 3.5 and 2.5 Ma which were near identically less radiogenic to the modern Arabian seawater. However, the intervening periods reflect more radiogenic ϵNd values similar to the 11.5 to 7.5 Ma period. The authors partitioned the ϵNd values between 7.5 and 2 Ma into two sources for which independent evidence is required (I am unsure whether it is possible to provide such data).

Validity: The validity of the manuscript's data is robust as the authors deftly dealt with the issues of boundary exchange, fossil fish teeth/debris, and detrital fraction to validate authigenic ϵNd values at the IODP Site U147.

Significance: The inferred "circulation pattern" proposed by the authors from the authigenic ϵNd values for late Miocene is very different compared to the late Pleistocene deep ocean circulation. However, to my surprise, the reconstructed ϵNd data from the eastern Indian Ocean appear consistent with the ϵNd data from the Atlantic Ocean. The coherency between the active (North Atlantic) and passive (Indian) basins should provide a cross-check to assess the modeling results (modelers are always excited about proposing storing carbon in the deep ocean); there lies one of the broader significances of the study.

Data and Methodology: The authors used established and time-tested methodology to acquire ϵNd data from the IODP Site U1457. It should be mentioned that the authors have corroborated the methods of extracting ϵNd data with the other published data from the northern Indian Ocean (Wilson et al., 2015; Piotrowski et al., 2009) as well as through their previous publications (Lathika et al., 2021; Rahman et al., 2020).

Analytical approach: Acceptable standard international analytical protocol used to acquire ϵNd data in the manuscript is currently used by the Nd-community.

Suggested improvements: The manuscript needs streamlining; this is NOT to cast an aspersion to the hypothesis or interpretation but to guide the authors to improve it further. There is a bit of reorganization, shortening of some sections and elaborating some sections (it is not a contradictory statement within the constrain of the length of the manuscript), modification of figures, and updating references.

The authors may consider adding three main points to their manuscript: (1) a brief discussion of the addition of the West Antarctic ice sheet as appears from the benthic oxygen isotopes; (b) Oxygen and carbon isotopes records in benthic foraminifera from the nearby DSDP/ODP Sites 214/216 and 757/758, respectively, in the northern Indian Ocean and eastern Arabian Sea such that circulation story (either matches or contradicts using the $\delta^{13}\text{C}$ data); and (c) the onset of the Indian (and Asian) monsoon using the planktonic foraminifera *Globigerinoides bulloides* and changes in vegetation from C3/C4 plants. Addition of this discussion at the expense of shortening the debate on ϵNd would provide an overarching story the authors seem to present but are currently in an incipient state.

Clarity and context: The manuscript text must be improved and improve the

contextualization (see above).

References: The authors need to widen the list of references by adopting divergent views (it does not mean that the authors need to accept those views).

Your expertise: I am comfortable in assessing the content of the manuscript with ease as I have familiarity with the ϵNd data through my publications from the northern Indian Ocean and the North Atlantic.

Reviewer #2 (Remarks to the Author):

This is a review of "Onset of modern-like deep water circulation in Indian Ocean caused by Central American Seaway closure" by Prabhat and colleagues for potential publication in Nature Communications.

In this manuscript, the authors present a new ϵNd record from the deep northwestern Indian Ocean spanning the interval between the Late Miocene (~ 11 Ma) and the Late Pliocene (~ 3 Ma). The authors suggest that the progressive excursion of ϵNd values toward less radiogenic values is driven by an increasing contribution from North Component Water (NCW, a proto-AMOC) and a decreasing contribution from Pacific Deep Water (PDW) via Antarctic Bottom Water (AABW) between 9 and 6 Ma. This change in the respective contribution of NCW and PDW to the deep water bathing the northwestern Indian Ocean is explained by the progressive closure of the Central American Seaway (CAS) across this time interval. Following this progressive closure, stable modern-like conditions would have been established after 6 Ma.

This hypothesis and the narrative build around it in the manuscript is appealing and new ϵNd data from the paleo Indian Ocean is certainly welcomed. However, there is a significant number of issues that need to be resolved to provide a satisfying demonstration of this story—so far, I am not at all convinced. Besides, there are confusing sentences and some misquoted citations.

I recommend to return the manuscript to the author with major revisions and I am willing to review a subsequent version provided that comments have been taken into account and that citations and figures have been fixed and the text stripped down for clarity.

I should also mention that as a paleoclimate modeler, I am in no position to meaningfully judge the experimental side of the work (described, e.g., in the Sample preparation and Nd isotopic analysis sections of the Methods) and I consider in the following that the authors employed an analytical procedure consistent with the state-of-the-art.

Major points

1. A good deal of the manuscript relies on the exclusion of a significant part of the reported ϵNd measurements—supposedly because these were influenced by terrestrial inputs obscuring changes in deep water provenance—, leading to nicely decreasing long-term ϵNd values at site U1457 that the authors interpret as reflecting increased influence of NCW and decreasing influence of PDW. The authors discard some points as affected by increased dust supply driven by episodes of enhanced aridity and stronger winds during the 7.4 – 6 Ma, ~ 3.5 Ma and 2.5 – 2 Ma intervals, as ascertained from other indicators (Fig. 3). These intervals also show low sedimentation rate suggesting reduced weathering fluxes from continental rivers and are periods of increased productivity as documented by increased TOC and TN contents. I am not saying that the mechanism inferred by the authors to explain the radiogenic "excursion" may not be possibly valid but I think there might be a simpler explanation. During times of lower (higher) sedimentation rate, detrital material probably makes for

a lower (higher) proportion of the total bulk sediment and consequently there are less (more) chances to incorporate dissolved detrital Nd during the leaching of bulk sediments. This would explain why the leachate signature is partially dragged toward the detrital value during times of high sedimentation rates. Interestingly, detrital eNd values during times of high sedimentation (11-9 Ma, 8.2-7.5 Ma, 6-5.6 Ma and 3.5-2.5 Ma) reproduce the decreasing trend seen in leachate values. Both the absolute rate of sedimentation and the detrital signature thus influence leachate signature during times of relatively high sedimentation. This hypothesis is consistent with less radiogenic leachate values between 8.2 and 7.5 Ma than between 11 and 9 Ma because, although detrital eNd carries the same signature, the 8.2-7.5 Ma sedimentation rate is roughly twice the 11-9 Ma. It is also consistent with even less radiogenic values between 6 and 5.6 Ma than between 8.2 and 7.5 Ma because now, although the sedimentation rate is similar, the detrital eNd signature is less radiogenic (Fig. 3).

I think the authors should test their results against this hypothesis and provide unambiguously arguments to demonstrate the validity of their story. At the moment, I don't necessarily buy the hypothetical dust argument to exclude half of the record.

2. I don't agree with the authors tossing out the Fe-Mn crust record simply on the basis of their extremely low growth rate. They argue that these crusts average the deep water signature over longer periods (which are how long exactly?) and thus do not capture variations related to changes in deep water circulation. If indeed there is a change in the deep water eNd signature because of a different provenance of water masses (Pacific vs Atlantic) between 9 and 6 Ma as suggested by the authors, then there should be a decreasing trend visible in the Fe-Mn records from the deep Indian Ocean, at least after 6 Ma. This is not the case in the SS 663 record even in the point dated after 3 Ma, that is, more than 3 million years after the inferred reorganization of water masses. In addition, the authors seem to have forgotten to include the 109-DC record from the Madagascar basin (O'Nions et al. 1998). This record is valuable because it is located upstream of site U1457 while the SS 663 record is located downstream regarding deep water pathways. Interestingly, neither of these records exhibit any change in the eNd signature of deep waters across the 11-3 Ma interval and this argues against the hypothesis proposed in this manuscript.

3. Assuming that leachate eNd values robustly reflect changes in deep water signature (but see comments above), I also have a problem with the link between the evolution of the contribution from Pacific waters vs NCW and the eNd record. In the Late Miocene, evidence point to a CAS open to intermediate depth at most and recent global paleogeographic reconstructions (He et al. 2021) therefore propose ~ 1400 m for the mid-Miocene (14 Ma) and ~ 200 m for the latest Miocene (6 Ma). This implies the CAS is closed to deep water since before the mid-Miocene so that recirculated Pacific Deep Waters (rPDW) reach the Southern Ocean and contribute to Lower Circumpolar Deep Water with AABW throughout the record presented in the manuscript.

With limited or no export of NCW to the Southern Ocean prior to 9 Ma, then the relative contribution of rPDW and AABW to LCDW can appropriately make for the radiogenic signal observed at site U1457 (but see Kirillova et al. 2019 and Poore et al. 2006 who suggest variable export of NCW to the Southern Ocean between 11.5 and 9.5 Ma, which is not observed in the eNd record presented in this manuscript).

However, I don't understand why the contribution of rPDW to LCDW would progressively disappear after 9 Ma (Fig. 5 and the author semi-quantitative estimates of water-mass mixing). It is true that increased contribution of NCW to LCDW should reduce the contribution of AABW and rPDW to it but there is no convincing argument to make rPDW disappear from the equation. As far as I understand it, the deep water pathway in the Pacific was not affected by the later stage of CAS closure (from intermediate to shallow depth) and the Drake Passage has been open to deep water flow since much earlier (Eagles and Jokat 2014). The semi-quantitative calculation should therefore consider 3 deep water masses at least from 9 Ma onwards (alternatively Dausmann et al. 2017 suggest that the eNd signature of CDW can be modeled as binary mixing between NCW and PDW).

4. Related to point 3., the Poore et al. 2006 NCW export to the Southern Ocean (reproduced by the authors on Fig. 4) shows significant variations of this export, in particular a decrease between 8 and 7 Ma. Would not this suggest that between 8 and 7 Ma, the deep water conditions were roughly the same as prior to 9 Ma? If so then what is the reason for the lower leachate ϵ_{Nd} values between 8 and 7.5 Ma relative to between 11 and 9 Ma?

In addition, I wonder how the authors explain the persistence of their inferred modern-like conditions after 6 Ma in a context of probably significant NCW (or NADW) variations during the latest Miocene and Pliocene (Dausmann et al. 2017).

Minor points

I. 23-25. This is a bit confusing. If there are no suitable records then how can significant GOC changes be inferred?

I. 26-27. Precise why.

I. 31. The exact closure date of the CAS is quite debated. Change to "caused by progressive shoaling of the Central American Seaway".

I. 50-51. Remove sentence. You do not mention orbits or greenhouse gas anymore in the manuscript.

I. 51-55. This sentence is confusing and hard to read.

I. 59. I don't understand the meaning of "it acts only as a host for the GOC".

I. 59. Located at the terminal end => located at one of the terminal end

I. 67-68. I don't agree (see comment above). You should see a long-term change.

I. 85-90. Yes, but you implicitly assume that major locations of deep water formation have not changed since the Late Miocene, only the contribution of deep water masses to the deep Indian Ocean. What about the suggestion that deep water formation existed in the North Pacific during the Pliocene (Burlis et al. 2017)?

I. 146 west-flowing => westward-flowing

I. 209-211. What about the influence of Pacific waters?

I. 225. Remove the portion after "if".

I. 235. Precise timescale of formation.

I. 248-249. How do you infer that it remained stable within the limit of Quaternary variability? If the radiogenic points are indeed affected by a terrestrial signature, there is almost no variability in your record.

I. 293-295. Maybe worth discussing the model results of Sentman et al. 2018 who show that the closure of the CAS has limited impact on AMOC strength in a state-of-the-art climate model. They also show that the closure of the CAS only marginally affects the contribution of NCW to the northwestern Indian Ocean.

I. 298. References 10 and 11 do not focus on the Atlantic Ocean or NCW production.

I. 300. Same comment about the references.

I. 346-347. I don't get the link between Late Miocene cooling events and the reduced influx of PW through constricting the CAS.

I. 472-475. This is flawed because the reduction in rPDW should be at least progressive. But see major comment above.

Figure 4. I cannot find the Billups et al. 2002 %NCW curve that is plotted. If you have recalculated it based on another figure from Billups et al. 2002, you should precise it. If not, then I don't know where this curve comes from.

References

- Billups, K., Channell, J. E. T., & Zachos, J. (2002). Late Oligocene to early Miocene geochronology and paleoceanography from the subantarctic South Atlantic. *Paleoceanography*, 17(1), 4-1.
- Burls, N. J., Fedorov, A. V., Sigman, D. M., Jaccard, S. L., Tiedemann, R., & Haug, G. H. (2017). Active Pacific meridional overturning circulation (PMOC) during the warm Pliocene. *Science advances*, 3(9), e1700156.
- Dausmann, V., Frank, M., Gutjahr, M., & Rickli, J. (2017). Glacial reduction of AMOC strength and long-term transition in weathering inputs into the Southern Ocean since the mid-Miocene: Evidence from radiogenic Nd and Hf isotopes. *Paleoceanography*, 32(3), 265-283.
- Eagles, G., & Jokat, W. (2014). Tectonic reconstructions for paleobathymetry in Drake Passage. *Tectonophysics*, 611, 28-50.
- He, Z., Zhang, Z., Guo, Z., Scotese, C. R., & Deng, C. (2021). Middle Miocene (~ 14 Ma) and Late Miocene (~ 6 Ma) Paleogeographic Boundary Conditions. *Paleoceanography and Paleoclimatology*, 36(11), e2021PA004298.
- Kirillova, V., Osborne, A. H., Störli, T., & Frank, M. (2019). Miocene restriction of the Pacific-North Atlantic throughflow strengthened Atlantic overturning circulation. *Nature communications*, 10(1), 1-7.
- O'Nions, R. K., Frank, M., von Blanckenburg, F., & Ling, H. F. (1998). Secular variation of Nd and Pb isotopes in ferromanganese crusts from the Atlantic, Indian and Pacific Oceans. *Earth and Planetary Science Letters*, 155(1-2), 15-28.
- Poore, H. R., Samworth, R., White, N. J., Jones, S. M., & McCave, I. N. (2006). Neogene overflow of northern component water at the Greenland-Scotland Ridge. *Geochemistry, Geophysics, Geosystems*, 7(6).
- Sentman, L. T., Dunne, J. P., Stouffer, R. J., Krasting, J. P., Toggweiler, J. R., & Broccoli, A. J. (2018). The mechanistic role of the central American seaway in a GFDL earth system model. Part 1: Impacts on global ocean mean state and circulation. *Paleoceanography and Paleoclimatology*, 33(7), 840-859.

Response To Reviewers Comments

Manuscript No.: NCOMMS-21-41188

Manuscript Title: Onset of modern-like deep water circulation in Indian Ocean caused by Central American Seaway closure.

REVIEWER COMMENTS

Reviewer #1 (Remarks to the Author):

Noteworthy results: The manuscript titled “Onset of modern-like deep water circulation in the Indian Ocean caused by Central American Seaway closure” by Prabhat et al. contains a lot of valuable data and provides new insights about the onset of deep ocean circulation using the neodymium radiogenic isotopes. It’s an impressive feat with the vast set of data authors provide to illuminate the resumption of the Miocene circulation on a tectonic scale. The authors accurately claim that there are NO high-resolution ϵ_{Nd} data in the Indian Ocean for the Miocene period compared to equivalent data in the Atlantic Ocean at three ODP Caribbean and one Blake Crust site. The authors placed their data into a broader context using published data mainly from the Atlantic Ocean to very convincingly argue for the 11.5 and 7.5 Ma period. One of the other interesting findings is the ϵ_{Nd} values between 6 and ~3.8 Ma and 3.5 and 2.5 Ma which were near identically less radiogenic to the modern Arabian seawater. However, the intervening periods reflect more radiogenic ϵ_{Nd} values similar to the 11.5 to 7.5 Ma period. The authors partitioned the ϵ_{Nd} values between 7.5 and 2 Ma into two sources for which independent evidence is required (I am unsure whether it is possible to provide such data).

Reply: We thank for the extensive reviews and constructive comments. We have addressed all the concerns and incorporated suggestions in the revised manuscript.

Validity: The validity of the manuscript’s data is robust as the authors deftly dealt with the issues of boundary exchange, fossil fish teeth/debris, and detrital fraction to validate authigenic ϵ_{Nd} values at the IODP Site U147.

Reply: We appreciate this comment.

Significance: The inferred “circulation pattern” proposed by the authors from the authigenic ϵ_{Nd} values for late Miocene is very different compared to the late Pleistocene deep ocean circulation. However, to my surprise, the reconstructed ϵ_{Nd} data from the eastern Indian Ocean appear consistent with the ϵ_{Nd} data from the Atlantic Ocean. The coherency between

the active (North Atlantic) and passive (Indian) basins should provide a cross-check to assess the modeling results (modelers are always excited about proposing storing carbon in the deep ocean); there lies one of the broader significances of the study.

Reply: We thank for the insightful comments.

Data and Methodology: The authors used established and time-tested methodology to acquire ϵ_{Nd} data from the IODP Site U1457. It should be mentioned that the authors have corroborated the methods of extracting ϵ_{Nd} data with the other published data from the northern Indian Ocean (Wilson et al., 2015; Piotrowski et al., 2009) as well as through their previous publications (Lathika et al., 2021; Rahman et al., 2020).

Reply: We thank and appreciate this comment.

Analytical approach: Acceptable standard international analytical protocol used to acquire ϵ_{Nd} data in the manuscript is currently used by the Nd- community.

Reply: Thanks for the comment.

Suggested improvements:

The manuscript needs streamlining; this is NOT to cast an aspersion to the hypothesis or interpretation but to guide the authors to improve it further. There is a bit of reorganization, shortening of some sections and elaborating some sections (it is not a contradictory statement within the constrain of the length of the manuscript), modification of figures, and updating references.

Reply: We thank for the constructive reviews and suggestions. We have made the changes in the text and figures.

The authors may consider adding three main points to their manuscript:

(1) a brief discussion of the addition of the West Antarctic ice sheet as appears from the benthic oxygen isotopes;

Reply: Thanks for the suggestion. In the revised manuscript, we have added a brief discussion on the West Antarctic ice sheet and its relation to Antarctic Bottom Water (AABW) formation (Line No. 221–228) and provided a supplementary figure (S6) on Antarctic Ice Sheet growth in the Cenozoic derived from benthic $\delta^{18}O$ record (Rohling et al., 2021; Westerhold et al., 2020).

(b) Oxygen and carbon isotopes records in benthic foraminifera from the nearby DSDP/ODP Sites 214/216 and 757/758, respectively, in the northern Indian Ocean and eastern Arabian Sea such that circulation story (either matches or contradicts using the $d_{13}C$ data); and

Reply: Thanks you for the suggestion. We have added discussion about the Indian Ocean and global ocean benthic $\delta^{13}\text{C}$ and $\delta^{18}\text{O}$ records and their link to the evolution of the water mass circulation in the Indian Ocean in the main text (Line no. 210–228) along with a new supplementary figure (Figure S5).

(c) the onset of the Indian (and Asian) monsoon using the planktonic foraminifera *Globigerinoides bulloides* and changes in vegetation from C3/C4 plants.

Reply: Thanks for the suggestion. In order to examine the role of Indian monsoon influencing authigenic ϵ_{Nd} record, we have presented the *Globigerina bulloides* record from the Oman coast ODP site 722 (Huang et al., 2007) and 730 (Gupta et al., 2015) (Fig 3a, b) and change in vegetation from C3 to C4 plants (Quade et al., 1989) (Fig 3c), (Line no. 181–185).

Addition of this discussion at the expense of shortening the debate on ϵ_{Nd} would provide an overarching story the authors seem to present but are currently in an incipient state.

Reply: Thanks for the suggestion. In view of this comments, we have thoroughly revised the text and added more discussion in support of our key findings and conclusion in the revised manuscript.

Clarity and context: The manuscript text must be improved and improve the contextualization (see above).

Reply: Thanks for the suggestions and comments. We have incorporated all the suggestions and thoroughly edited the text to improve the manuscripts.

References: The authors need to widen the list of references by adopting divergent views (it does not mean that the authors need to accept those views).

Reply: Thanks for the suggestions. We have discussed the views and prevailing hypothesis in the relevant context and cited appropriate references wherever necessary.

Your expertise: I am comfortable in assessing the content of the manuscript with ease as I have familiarity with the ϵ_{Nd} data through my publications from the northern Indian Ocean and the North Atlantic.

Reviewer #2 (Remarks to the Author):

This is a review of “Onset of modern-like deep water circulation in Indian Ocean caused by Central American Seaway closure” by Prabhat and colleagues for potential publication in Nature Communications.

In this manuscript, the authors present a new ϵ_{Nd} record from the deep northwestern Indian Ocean spanning the interval between the Late Miocene (~ 11 Ma) and the Late Pliocene (~ 3 Ma). The authors suggest that the progressive excursion of ϵ_{Nd} values toward less radiogenic values is driven by an increasing contribution from North Component Water (NCW, a proto-AMOC) and a decreasing contribution from Pacific Deep Water (PDW) via Antarctic Bottom Water (AABW) between 9 and 6 Ma. This change in the respective contribution of NCW and PDW to the deep water bathing the Indian Ocean is explained by the progressive closure of the Central American Seaway (CAS) across this time interval. Following this progressive closure, stable modern-like conditions would have been established after 6 Ma.

This hypothesis and the narrative build around it in the manuscript is appealing and new ϵ_{Nd} data from the paleo Indian Ocean is certainly welcomed. However, there is a significant number of issues that need to be resolved to provide a satisfying demonstration of this story—so far, I am not at all convinced. Besides, there are confusing sentences and some misquoted citations.

I recommend to return the manuscript to the author with major revisions and I am willing to review a subsequent version provided that comments have been taken into account and that citations and figures have been fixed and the text stripped down for clarity. I should also mention that as a paleoclimate modeler, I am in no position to meaningfully judge the experimental side of the work (described, e.g., in the Sample preparation and Nd isotopic analysis sections of the Methods) and I consider in the following that the authors employed an analytical procedure consistent with the state-of-the-art.

Major points

1. A good deal of the manuscript relies on the exclusion of a significant part of the reported ϵ_{Nd} measurements—supposedly because these were influenced by terrestrial inputs obscuring changes in deep water provenance—, leading to nicely decreasing long-term ϵ_{Nd} values at site U1457 that the authors interpret as reflecting increased influence of NCW and decreasing influence of PDW. The authors discard some points as affected by increased dust supply driven by episodes of enhanced aridity and stronger winds during the 7.4 – 6 Ma, ~ 3.5 Ma and 2.5 – 2 Ma intervals, as ascertained from other indicators (Fig. 3). These

intervals also show low sedimentation rate suggesting reduced weathering fluxes from continental rivers and are periods of increased productivity as documented by increased TOC and TN contents. I am not saying that the mechanism inferred by the authors to explain the radiogenic “excursion” may not be possibly valid but I think there might be a simpler explanation.

During times of lower (higher) sedimentation rate, detrital material probably makes for a lower (higher) proportion of the total bulk sediment and consequently there are less (more) chances to incorporate dissolved detrital Nd during the leaching of bulk sediments. This would explain why the leachate signature is partially dragged toward the detrital value during times of high sedimentation rates. Interestingly, detrital ϵ_{Nd} values during times of high sedimentation (11-9 Ma, 8.2-7.5 Ma, 6-5.6 Ma and 3.5-2.5 Ma) reproduce the decreasing trend seen in leachate values. Both the absolute rate of sedimentation and the detrital signature thus influence leachate signature during times of relatively high sedimentation. This hypothesis is consistent with less radiogenic leachate values between 8.2 and 7.5 Ma than between 11 and 9 Ma because, although detrital ϵ_{Nd} carries the same signature, the 8.2-7.5 Ma sedimentation rate is roughly twice the 11-9 Ma. It is also consistent with even less radiogenic values between 6 and 5.6 Ma than between 8.2 and 7.5 Ma because now, although the sedimentation rate is similar, the detrital ϵ_{Nd} signature is less radiogenic (Fig.3). I think the authors should test their results against this hypothesis and provide unambiguously arguments to demonstrate the validity of their story. At the moment, I don't necessarily buy the hypothetical dust argument to exclude half of the record.

Reply: We thank the reviewer for insightful comments and suggestions.

To address the issue of partial/selective leaching of detrital Nd, we measured ϵ_{Nd} in the fish teeth samples from various core depths (n=11, Fig. 2a.). Fish teeth/debris ϵ_{Nd} is considered to be more robust and reliable archive to reconstruct the water mass signature without detrital contamination (Frank, 2002; Martin and Haley, 2000). The agreement between the ϵ_{Nd} values of the fish teeth and leachate samples (Figure 2a, Supplementary Figures S2) gives us more confidence that the ϵ_{Nd} values of the leachate successfully record the past deep water signature without significant influence of partial leaching of detrital Nd (Line no. 129–137). Due to the restricted occurrence of the fish teeth and the foraminifera in the present core, Nd isotope was measured in the sediment leach fraction to generate a high-resolution continuous record. However, multiple evidences such as slower sedimentation rates, increased TOC and TN contents, indicate that radiogenic excursions resulted from the enhanced dust supply during the arid periods and dust become the dominant fractions of the bulk sediments. Therefore, leaching of these sediments will drag the ϵ_{Nd} values towards the detrital signature as Fe-Mn oxyhydroxides coatings was not in equilibrium with the deep-

water Nd composition, and hence we refrain from interpreting these three radiogenic excursions in terms of deep water circulation (Line no. 158–201).

2. I don't agree with the authors tossing out the Fe-Mn crust record simply on the basis of their extremely low growth rate. They argue that these crusts average the deep water signature over longer periods (which are how long exactly?) and thus do not capture variations related to changes in deep water circulation. If indeed there is a change in the deep water ϵ_{Nd} signature because of a different provenance of water masses (Pacific vs Atlantic) between 9 and 6 Ma as suggested by the authors, then there should be a decreasing trend visible in the Fe-Mn records from the deep Indian Ocean, at least after 6 Ma. This is not the case in the SS 663 record even in the point dated after 3 Ma, that is, more than 3 million years after the inferred reorganization of water masses. In addition, the authors seem to have forgotten to include the 109-DC record from the Madagascar basin (O'Nions et al. 1998). This record is valuable because it is located upstream of site U1457 while the SS 663 record is located downstream regarding deep water pathways. Interestingly, neither of these records exhibit any change in the ϵ_{Nd} signature of deep waters across the 11-3 Ma interval and this argues against the hypothesis proposed in this manuscript.

Reply: Thanks for the comments and suggestions. In the revised version we have removed the line "Fe-Mn crust average the deep water signature over longer periods" and modified text. We have added the crust 109-DC record from the Madagascar basin (Line no. 236) and provided a new figure in the supplementary (Figure S2) to show the trend in the compiled deep water Fe-Mn crust ϵ_{Nd} records in the revised version.

Fe-Mn Crust Name	Water Depth (m)	Location (Lat., Long.)	Growth Rate (mm/Ma)	Resolution (Ma)	References
DODO 232D	4119	5°23.0'S, 97°29.0'E	4.31	3.2	Frank et al. (2006)
SS-663	5300	13°S, 76°E	2.82	2	O'Nions et al. (1998)
109 D-C	5700	28°S, 61°E	1.55	2.66	

All the crust records compiled here are from deeper depths (>4000 m) than our core site (3500 m). The depth below 3800 m in the Indian Ocean is mostly occupied by the pure Antarctic bottom water (Mantyla and Reid, 1995) (AABW, $\epsilon_{Nd} = -8 \pm 1$) in the modern ocean. However, the earlier studies note that AABW has been the densest water mass since the

middle Miocene (Flower and Kennett, 1995) and occupied the bottom of all ocean basins. Thus, these Fe-Mn crusts record the ϵ_{Nd} variability of AABW and doesn't get affected by the changes in the formation and export of NCW and PDW. However, our core site is situated at a shallower depth than the Fe-Mn crusts and bathed by a mixture of AABW and NADW in the modern ocean (Dileep Kumar and Li, 1996; Goswami et al., 2012; Mantyla and Reid, 1995). Thus, our authigenic ϵ_{Nd} records the deep water signature and reflects the variability and changes in the mixing proportion of these water masses since the late Miocene.

3. Assuming that leachate ϵ_{Nd} values robustly reflect changes in deep water signature (but see comments above), I also have a problem with the link between the evolution of the contribution from Pacific waters vs NCW and the ϵ_{Nd} record. In the Late Miocene, evidence point to a CAS open to intermediate depth at most and recent global paleogeographic reconstructions (He et al. 2021) therefore propose ~ 1400 m for the mid-Miocene (14 Ma) and ~ 200 m for the latest Miocene (6 Ma). This implies the CAS is closed to deep water since before the mid-Miocene so that recirculated Pacific Deep Waters (rPDW) reach the Southern Ocean and contribute to Lower Circumpolar Deep Water with AABW throughout the record presented in the manuscript. With limited or no export of NCW to the Southern Ocean prior to 9 Ma, then the relative contribution of rPDW and AABW to LCDW can appropriately make for the radiogenic signal observed at site U1457 (but see Kirillova et al. 2019 and Poore et al. 2006 who suggest variable export of NCW to the Southern Ocean between 11.5 and 9.5 Ma, which is not observed in the ϵ_{Nd} record presented in this manuscript). However, I don't understand why the contribution of rPDW to LCDW would progressively disappear after 9 Ma (Fig. 5 and the author semi-quantitative estimates of water-mass mixing). It is true that increased contribution of NCW to LCDW should reduce the contribution of AABW and rPDW to it but there is no convincing argument to make rPDW disappear from the equation. As far as I understand it, the deep water pathway in the Pacific was not affected by the later stage of CAS closure (from intermediate to shallow depth) and the Drake Passage has been open to deep water flow since much earlier (Eagles and Jokat

2014). The semi-quantitative calculation should therefore consider 3 deep water masses at least from 9 Ma onwards (alternatively Dausmann et al. 2017 suggest that the eNd signature of CDW can be modeled as binary mixing between NCW and PDW).

Reply: We thank for this insightful comment. As per the comments and suggestions, in the revised manuscript, we have made necessary changes in the main text (Line no. 316–329) and Figure 5 (schematic and mechanism) for better clarity in our explanations.

The primary concern raised here is the disappearance of the rPDW contributions to LCDW after ~9 Ma. Further, the reviewer has suggested considering either a three-endmember mixing or NCW and PDW as the end member for semi-quantitative mixing calculations.

Prior to ~9 Ma, when the export of NCW to the Southern Ocean was limited due to weaker AMOC, the rPDW was the part of the LCDW (mixture of AABW and rPDW) (Figure 5d and Figure 'a' shown in below panel). After 9 Ma, due to the progressive closure of CAS, export of Pacific water into the north Atlantic got reduced and resulted in strengthening of NCW formation and its export to the Southern Ocean. Thus, the rPDW in the lower cell of CDW got replaced by the NCW and the rPDW became the part of the upper cell of the CDW (UCDW) due to its lower density than NCW (Figure 5e and Figure 'b' shown in below panel). In the modern day water circulation, rPDW mixes with other upwelled deep water masses to form the upper cell of circumpolar deep water (UCDW), as illustrated in the figure b below. Thus, rPDW did not disappear from the Southern Ocean circulation system, rather it changed its vertical position (shoaled to shallower depths) after 9 Ma as illustrated in the panels below and Figure 5d, e (main text).

Figure a). Deep water circulation cell before ~9 Ma, when the NCW export to Southern Ocean was weak; b). deep water circulation cell since ~9 Ma after enhanced export of NCW to Southern Ocean. Figure b illustrates the condition similar to the modern day deep water circulation cell.

Dausmann et al. (2017) used NCW and PDW as endmembers in the binary mixing calculation for site 1088 because this site is located within the mixing zone of NADW and UCDW (mixture of NADW and PDW). Whereas our core site is bathed by the LCDW with a mixture of modified NADW (15 -10%) and AABW (85-90%) (Goswami et al., 2014; Lathika et

al., 2021). Hence, we have used the NCW and AABW as end member in the binary mixing calculation for the specific time interval (~8 Ma onwards) when NCW export was enhanced and resulted in less radiogenic ϵ_{Nd} values. During the period of weaker NCW export to Southern Ocean (before ~9 Ma), our site was mainly bathed by LCDW with a mixture of AABW and rPDW. Hence, before 9 Ma, we have used AABW and PDW as end members in the binary mixing model.

4. Related to point 3., the Poore et al. 2006 NCW export to the Southern Ocean (reproduced by the authors on Fig. 4) shows significant variations of this export, in particular a decrease between 8 and 7 Ma. Would not this suggest that between 8 and 7 Ma, the deep water conditions were roughly the same as prior to 9 Ma? If so then what is the reason for the lower leachate ϵ_{Nd} values between 8 and 7.5 Ma relative to between 11 and 9 Ma? In addition, I wonder how the authors explain the persistence of their inferred modern-like conditions after 6 Ma in a context of probably significant NCW (or NADW) variations during the latest Miocene and Pliocene (Dausmann et al. 2017).

Reply: Thanks for the comment.

Although in their study Poore et al. (2006) have assumed that the mixing zone between UCDW and NCW has remained constant over the entire period, however, they agree that geometry of mixing (geographical position and mixing zone) might have changed through the time and this may impact the %NCW estimation and the conclusion. The sensitivity in %NCW estimation with respect to the geometry of mixing is noted by Poore et al. (2006) with the inclusion of $\delta^{13}C$ data from site 704 which is further south (in comparison to sites 360 and 1088) and the results indicated a peak in %NCW at ~7 Ma. The %NCW curve without inclusion of record from ODP 704 and with inclusion of ODP 704 has been plotted in the following figure a and b respectively. This implies that changes in the density at deeper depths with the enhanced formation and southward export of the NCW may affect the estimation of %NCW.

Changes in geometry of mixing could result from the enhanced southward export of the NCW, higher latitude cooling during the late Miocene (Figure 4b, main text) and gradual cooling of the bottom water (Figure 4c, main text) and may result in the discrepancy of the estimation of water mass depending on the depth and geographical position of the study sites.

Figure a). %NCW estimation without $\delta^{13}\text{C}$ data from core 704; b). %NCW estimation after incorporating $\delta^{13}\text{C}$ data from core 704. Green band highlights the variation in %NCW contribution with addition of site 704 due to possible change in the geometry of mixing.

Poore et al. (2006) have interpreted the rapid increase in the %NADW during the interval of ~6 Ma to 2.7 Ma in terms of the NCW production rather than that of the proportional changes in the mixing proportion of the water masses. Similarly, Billups et al. (2002) have reported weaker NCW at the site 1088 during most of the late Miocene interval and suggests enhanced NCW formation and export to Southern Ocean after ~6.5 Ma.

It is important to note that site 1088 (~2082 m) and the other sites mentioned in the Poore et al. (2006) belongs to shallower depth in the mixing zone of UCDW and the NCW, and hence don't record the deep water mixing at deeper depths similar to our study site U1457 (~3522 m) being bathed by the mixture of modern day AABW and NCW. Thus, the variability in %NCW can be episodic due to variable mixing proportion of the NCW and the southern sourced water masses. These variations were not observed in our record possibly due to deeper depth of our study site.

The NCW production was strongest during the Pliocene, however, our record could not resolve this due to coarse resolution of samples during the Pliocene and could not capture all the variability similar to shallower site of 1088 from Dausmann et al. (2017). Also, there could be another possibility that even though NCW formation increased, the mixing geometry between NCW and AABW remained similar to that of the modern day due to their similar density gradient, resulting in modern-like ϵ_{Nd} values.

Minor points

. 23-25. This is a bit confusing. If there are no suitable records then how can significant GOC changes be inferred?

Reply: Thanks for the comment. This sentence is modified in the revised manuscript (Line no. 21–24).

. 26-27. Precise why.

Reply: Thanks for the comment. Since the Indian Ocean does not have its own deep water formation region, it is ventilated only from the southern side by the NCW and AABW. Hence, any changes in the deep water) feeding to Indian Ocean would reflect to changes in the source region or changes that took place along the flow path of the deep water circulation. This sentence is removed, and text has been modified in the abstract.

. 31. The exact closure date of the CAS is quite debated. Change to “caused by progressive shoaling of the Central American Seaway”.

Reply: Thanks for the suggestion. This sentence is modified accordingly (Line no. 30-31).

. 50-51. Remove sentence. You do not mention orbits or greenhouse gas anymore in the manuscript.

Reply: Thanks for the suggestion. Sentence has been deleted in the revised manuscript.

. 51-55. This sentence is confusing and hard to read.

Reply: Sentence has been modified for clarity (Line no. 50–53).

. 59. I don't understand the meaning of “it acts only as a host for the GOC”.

Reply: Sentence modified to “it acts only as a host for the deep water circulation (NCW and AABW)” (Line no. 57–58).

. 59. Located at the terminal end => located at one of the terminal end

Reply: Thanks. Sentence modified accordingly (Line no. 59).

. 67-68. I don't agree (see comment above). You should see a long-term change.

Reply: Thanks for the comment. In view of the major comment, we have thoroughly revised the discussion in the main text and supplementary about this aspect (Line no. 67-69 and 236-24) and also replied in the major comment No. 2.

. 85-90. Yes, but you implicitly assume that major locations of deep water formation have not changed since the Late Miocene, only the contribution of deep water masses to the deep Indian Ocean. What about the suggestion that deep water formation existed in the North Pacific during the Pliocene (Burls et al. 2017)?

Reply: Yes, we have not considered the deep water formation in the North Pacific suggested by Burls et al. (2017). Since it was less dense than the PDW/NCW and upwelled to shallower depth after crossing the equator did not affect our study site in the presence of the NCW. When the deep water formed in North Pacific (NPDW) moved southward it may get mixed with upwelling PDW to form the part of the UCDW. Since, NPDW formed in the North Pacific, theoretically it should have the radiogenic ϵ_{Nd} values similar to that of the PDW and can be considered as a part of rPDW.

. 146 west-flowing => westward-flowing

Reply: Thanks for the suggestion. This sentence is modified (Line no. 144).

. 209-211. What about the influence of Pacific waters?

Reply: Thanks for the comment. The Pacific water is characterised by a ϵ_{Nd} value of -3.5 ± 0.5 (Howe et al., 2016) and lower density water than the AABW. Hence, at the depth of study site U1457 (3522 m), it is not feasible for the Pacific water to occupy this depth completely (100% Pacific water).

As per the reviewer 1 suggestion regarding the reorganisation of the text, sentence is removed from the revised version.

. 225. Remove the portion after "if".

Reply: Thanks. This sentence is removed in the revised version.

. 235. Precise timescale of formation.

Reply: Thank you for the suggestion. Since sentence has been removed in the revised manuscript, we have provided the rate of formation for the Fe-Mn crust (i.e., 1 to 4.3 mm/Ma) from the Indian Ocean in the line no. 67 of main text. Details of each crust is provided in response of major comment 2.

. 248-249. How do you infer that it remained stable within the limit of Quaternary variability? If the radiogenic points are indeed affected by a terrestrial signature, there is almost no variability in your record.

Reply: Thanks for the comment. Our inference is based on record that the ϵ_{Nd} values of the deep water mass stayed within the ϵ_{Nd} range of -6 to -9 reported for the Quaternary glacial-interglacial periods (Lathika et al., 2021) and does not show high radiogenic values (-4.5 to -5.5). Since the situation after 6 Ma were similar to the interglacial/Holocene and NCW was stronger, hence ϵ_{Nd} values remained less radiogenic (~-9) without much variability.

. 293-295. Maybe worth discussing the model results of Sentman et al. 2018 who show that the closure of the CAS has limited impact on AMOC strength in a state-of-the-art climate model. They also show that the closure of the CAS only marginally affects the contribution of NCW to the northwestern Indian Ocean.

Reply: Thanks for the comments. Sentman et al. (2018) have suggested Pacific water influx through CAS during the open CAS scenarios (either deep or shallow sill with narrow or wide passage) exported into the South Atlantic rather than North Atlantic and did not contribute to AMOC strength. However, other modelling studies (Butzin et al., 2011; Nisancioglu et al., 2003) have suggested significant impact of CAS on the AMOC strength. These models based simulations produced divergent results and therefore highlight the need of independent proxy based reconstructions. The radiogenic ϵ_{Nd} records from the Blake Crust and ODP 998 (Figure 4a) demonstrates the export of the Pacific water to the North Atlantic which gradually became less radiogenic with the progressive closure of CAS. Hence, the northward export of the Pacific water through open CAS impacted the formation and export of northern component water, which is very much consistent with our findings and thus indicating significant impact of CAS on the AMOC and large scale deep water circulation.

. 298. References 10 and 11 do not focus on the Atlantic Ocean or NCW production.

Reply: Thanks for the suggestion. These references are removed, and appropriate references are cited here in the revised version (Line no. 288).

. 300. Same comment about the references.

Reply: Thanks. These references are removed, and suitable references are cited here in the revised manuscript (Line no. 291).

. 346-347. I don't get the link between Late Miocene cooling events and the reduced influx of PW through constricting the CAS.

Reply: Thanks for the comment. We agree that there is no direct connection between late Miocene cooling and reduced influx of PW due to the closure of CAS. We have modified the discussion in the revised version (Line no. 305–310).

Constriction of the CAS caused the reduced influx of fresh PW and resulted in the increased sea water salinity in the Atlantic and thus, stronger AMOC and NCW formation. The high latitude cooling of sea surface temperature during Late Miocene cooling may add to NCW formation and increased density of NCW through increased brine rejection. Therefore, late Miocene cooling might add to production of NCW along with stronger AMOC caused by the constriction of CAS.

. 472-475. This is flawed because the reduction in rPDW should be at least progressive. But see major comment above.

Reply: Thank you for the comment (Line No. 458–461). Yes, we agree that the reduction of rPDW should be gradual with the progressive closure of CAS. However, our record could not reflect this gradual change in rPDW due to coarser resolution of samples and a hiatus (0.6 Ma) after 9 Ma (Pandey et al., 2016).

Figure 4. I cannot find the Billups et al. 2002 %NCW curve that is plotted. If you have recalculated it based on another figure from Billups et al. 2002, you should precise it. If not, then I don't know where this curve comes from.

Reply: Thank you for the comment. The %NCW curve plotted in the figure 4d is taken from Figure 6D of Billups et al. (2002)

References

Billups, K., Channell, J.E.T., Zachos, J., 2002. Late Oligocene to early Miocene geochronology and paleoceanography from the subantarctic South Atlantic. *Paleoceanography* 17, 4-1-4-11.

Burls, N.J., Fedorov, A.V., Sigman, D.M., Jaccard, S.L., Tiedemann, R., Haug, G.H., 2017. Active Pacific meridional overturning circulation (PMOC) during the warm Pliocene. *3*, e1700156.

Butzin, M., Lohmann, G., Bickert, T., 2011. Miocene ocean circulation inferred from marine carbon cycle modeling combined with benthic isotope records. *Paleoceanography* 26.

Dausmann, V., Frank, M., Gutjahr, M., Rickli, J., 2017. Glacial reduction of AMOC strength and long-term transition in weathering inputs into the Southern Ocean since the mid-Miocene: Evidence from radiogenic Nd and Hf isotopes. *Paleoceanography* 32, 265-283.

Dileep Kumar, M., Li, Y.-H., 1996. Spreading of water masses and regeneration of silica and ^{226}Ra in the Indian Ocean. *Deep Sea Research Part II: Topical Studies in Oceanography* 43, 83-110.

Flower, B.P., Kennett, J.P., 1995. Middle Miocene deepwater paleoceanography in the southwest Pacific: Relations with East Antarctic Ice Sheet development. *10*, 1095-1112.

Frank, M., 2002. RADIOGENIC ISOTOPES: TRACERS OF PAST OCEAN CIRCULATION AND EROSIONAL INPUT. *Reviews of Geophysics* 40, 1-1-1-38.

Frank, M., Whiteley, N., van de Flierdt, T., Reynolds, B.C., O'Nions, K., 2006. Nd and Pb isotope evolution of deep water masses in the eastern Indian Ocean during the past 33 Myr. *Chemical Geology* 226, 264-279.

Goswami, V., Singh, S.K., Bhushan, R., 2014. Impact of water mass mixing and dust deposition on Nd concentration and ϵ Nd of the Arabian Sea water column. *Geochimica et Cosmochimica Acta* 145, 30-49.

Goswami, V., Singh, S.K., Bhushan, R., Rai, V.K., 2012. Temporal variations in $^{87}\text{Sr}/^{86}\text{Sr}$ and ϵ Nd in sediments of the southeastern Arabian Sea: Impact of monsoon and surface water circulation. *Geochemistry, Geophysics, Geosystems* 13, n/a-n/a.

Gupta, A.K., Yuvaraja, A., Prakasam, M., Clemens, S.C., Velu, A., 2015. Evolution of the South Asian monsoon wind system since the late Middle Miocene. *Palaeogeography, Palaeoclimatology, Palaeoecology* 438, 160-167.

Howe, J.N., Piotrowski, A.M., Noble, T.L., Mulitza, S., Chiessi, C.M., Bayon, G., 2016. North Atlantic Deep Water Production during the Last Glacial Maximum. *Nature communications* 7, 11765.

Huang, Y., Clemens, S.C., Liu, W., Wang, Y., Prell, W.L., 2007. Large-scale hydrological change drove the late Miocene C4 plant expansion in the Himalayan foreland and Arabian Peninsula. *Geology* 35, 531.

Lathika, N., Rahaman, W., Tarique, M., Gandhi, N., Kumar, A., Thamban, M., 2021. Deep water circulation in the Arabian Sea during the last glacial cycle: Implications for paleo-redox condition, carbon sink and atmospheric CO₂ variability. *Quaternary Science Reviews* 257, 106853.

Mantyla, A.W., Reid, J.L., 1995. On the origins of deep and bottom waters of the Indian Ocean. *Journal of Geophysical Research: Oceans* 100, 2417-2439.

Martin, E.E., Haley, B.A., 2000. Fossil fish teeth as proxies for seawater Sr and Nd isotopes. *Geochimica et Cosmochimica Acta* 64, 835-847.

Nisancioglu, K.H., Raymo, M.E., Stone, P.H., 2003. Reorganization of Miocene deep water circulation in response to the shoaling of the Central American Seaway. *Paleoceanography* 18.

O'Nions, R.K., Frank, M., von Blanckenburg, F., Ling, H.F., 1998. Secular variation of Nd and Pb isotopes in ferromanganese crusts from the Atlantic, Indian and Pacific Oceans. *Earth and Planetary Science Letters* 155, 15-28.

Pandey, D.K., Clift, P.D., Kulhanek, D.K., Scientists, a.t.E., 2016. Site U1457. *Proceedings of the International Ocean Discovery Program* 355, 1-49.

Poore, H.R., Samworth, R., White, N.J., Jones, S.M., McCave, I.N., 2006. Neogene overflow of Northern Component Water at the Greenland-Scotland Ridge. *Geochemistry, Geophysics, Geosystems* 7, n/a-n/a.

Quade, J., Cerling, T.E., Bowman, J.R., 1989. Development of Asian monsoon revealed by marked ecological shift during the latest Miocene in northern Pakistan. *Nature* 342, 163-166.

Rohling, E.J., Yu, J., Heslop, D., Foster, G.L., Opdyke, B., Roberts, A.P., 2021. Sea level and deep-sea temperature reconstructions suggest quasi-stable states and critical transitions over the past 40 million years. *7*, eabf5326.

Sentman, L.T., Dunne, J.P., Stouffer, R.J., Krasting, J.P., Toggweiler, J.R., Broccoli, A.J., 2018. The Mechanistic Role of the Central American Seaway in a GFDL Earth System Model. Part 1: Impacts on Global Ocean Mean State and Circulation. *33*, 840-859.

Westerhold, T., Marwan, N., Drury, A.J., Liebrand, D., Agnini, C., Anagnostou, E., Barnet, J.S.K., Bohaty, S.M., De Vleeschouwer, D., Florindo, F., Frederichs, T., Hodell, D.A., Holbourn, A.E., Kroon, D., Laurentano, V., Littler, K., Lourens, L.J., Lyle, M., Pälike, H., Röhl, U., Tian, J., Wilkens, R.H., Wilson, P.A., Zachos, J.C., 2020. An astronomically dated record of Earth's climate and its predictability over the last 66 million years. *Science* 369, 1383.

Reviewer #1 (Remarks to the Author):

The revised manuscript titled "Onset of modern-like deep water circulation in the Indian Ocean caused by Central American Seaway closure" by Prabhat et al. has addressed most of my concerns and questions. The authors have also improved the text as appeared from the track-changes document, which has significantly improved the principal tenor of the manuscript, which the authors have been struggling to convey.

One of the essential findings by Prabhat et al. is the demonstration of the integrity of the eNd values as a reliable deep-water circulation proxy proximal to the continental margin. The authors provided data on simultaneous extraction and measurements of eNd in fossil fish teeth, detrital and authigenic fractions, not an easy task or effort to overlook, primarily if terrestrial sediments were also deposited at the sediment core sites. For example, Naik et al. (2019 G3; their Figs. 2 and 4) demonstrated that the fossil fish teeth/broken bone fragments and foraminifers show insignificant differences in eNd values from a sediment core in the Bay of Bengal. Further, Naik et al. (2019) has demonstrated (their Fig. 4) insignificant eNd variability regardless of cleaning protocols such as cleaned foraminifers, decarbonated sediment leachates, acidic acid leachates, and uncleaned foraminifers being used. These earlier findings further support the validity of eNd extraction (Fig. S3) and hence, the eNd data of Lathika et al. (2021), one of the coauthors in Prabhat et al., as well as the current manuscript by Prabhat et al.

I appreciate authors' elaboration of the discussion using *G. bulloides* (surface ocean proxy), but it reflects the strength of the summer monsoon and eNd (bottom water proxy, at the IODP site U1457) and how the onset of the circulation was coupled to the intensity of the Indian summer monsoon. This description was dormant, but the revised manuscript clarifies the linkage.

I am not fully onboard with their End-member calculation method of apportioning eNd values to the AABW and NADW as I raised it to their initial submission of the manuscript; however, my hesitancy should not prevent the manuscript from being published should Nat. Comm. decides to accept it.

I have edited the manuscript's abstract (see attached), but I suppose either Nat. Comm or someone else could improve the text further.

Reviewer #1 Attachment on the following page

21 Abstract

22 Global overturning circulation underwent significant changes in the late Miocene, driven
23 by tectonic forcing, and impacted the global climate. Prevailing hypotheses related to the
24 deep water circulation (DWC) changes driven by the closure of the Central American
25 Seaways (CAS) and its widespread impact remains untested due to the paucity of suitable
26 records away from the CAS region. Here, we test the hypothesis of the large-scale
27 circulation changes by providing a high-resolution record of DWC since the late Miocene
28 (11.3 to ~2 Ma) from the NW Indian Ocean. Our investigation reveals a progressive shift
29 from Pacific-dominated DWC before ~9.0 Ma to the onset of a modern-like DWC system
30 in the Indian Ocean with Antarctic bottom water and northern component water at the
31 Miocene-Pliocene transition (~6 Ma) caused by progressive shoaling of CAS and
32 suggests a widespread impact.

Reviewer #2 (Remarks to the Author):

I appreciate the effort made by the authors to reply to each comment of the reviews. However, I regret to say that some responses to the major comments are in my opinion not satisfying and the new version of the manuscript pertaining to these comments is nowhere more convincing than the original.

Though I do not doubt the quality of the data presented in this new deep-sea record or reject the possibility that the authors' interpretation could be correct, I think the arguments presented in the manuscript are just too ambiguous or circular yet to firmly back up the authors' story. My recommendation is therefore to reject the manuscript in its current form.

1. I maintain my first major comment of the last round of reviews, given that the authors' answer simply repeats the argumentation presented in the manuscript and do not provide any other analysis that could help refute the alternative interpretation that I suggested. Again, I do not claim that I am correct but just that it is a possibility that should be dismissed and the authors have up to now failed to do so.

In particular, the authors answer states that "Fish teeth/debris ϵNd is considered to be more robust and reliable archive to reconstruct the water mass signature without detrital contamination (Frank, 2002; Martin and Haley, 2000)". But precisely, if fish teeth ϵNd is a reliable seawater signature not affected by detrital contamination, then how do the authors explain that fish teeth ϵNd is remarkably constant across the record? This instead suggests that the deep seawater ϵNd at this site remained remarkably similar between 11 and 2 Ma—note that this does not mean that deep sea currents did not change, only that any "new" current carried almost the same ϵNd signature.

To be fair, I follow the authors that detrital ϵNd was affected by competition between dust inputs (more radiogenic) and riverine inputs (less radiogenic) from westward flowing Himalayan rivers, the ϵNd signature of which decrease between 10 – 2 Ma (Carter et al. 2020). I do not contest their interpretation of increased aridification at 7.4 – 6 Ma, ~ 3.5 Ma and 2.5 – 2 Ma, leading to dust-dominated intervals in the detrital record that exhibit radiogenic excursions (as the authors show, these dust-dominated periods are coeval with low sedimentation rate suggesting a decreased riverine flux and thus a more radiogenic ϵNd signature in the detrital record).

My view diverges in that I suggest that at times of low sedimentation rate (i.e. the dust-dominated intervals excluded by the authors), which are incidentally also times of higher productivity (most likely marine but this is unclear), the proportion of detrital material incorporated in the bulk sediment decreases. Therefore, the authigenic ϵNd signature obtained from the bulk sediment is less likely to be affected by partial dissolution of detrital material during the leaching, and thus the authigenic ϵNd signature gets closer to the seawater signature. This is consistent with the fact that authigenic ϵNd and fish teeth ϵNd are similar during these intervals. Conversely, at times of high sedimentation rate (for which there are unfortunately no fish teeth ϵNd data), the detrital material makes up for a much larger proportion of the bulk sediment and the authigenic ϵNd is more likely affected by partial dissolution of detrital material. This acts to draw the authigenic ϵNd signature closer to the detrital signature and explains the decrease seen in the authigenic ϵNd record.

In summary, the authors would like to exclude the dust-dominated intervals at 7.4 – 6 Ma, ~ 3.5 Ma and 2.5 – 2 Ma because of potential terrestrial contamination, whereas I suggest that these intervals instead better record the deep sea ϵNd signature.

2. In the response to my other major points, I have a problem with the fact that the authors invoke the modern ocean deep water configuration to justify what they observe (or calculate in the case of the binary mixing model) in the late Miocene. The issue is that assuming that modern vertical water mass distribution is the same in the Miocene is somewhat a circular reasoning if the aim is to document deep water evolution towards modern conditions. The vertical distribution of water mass is not even known for sure at the Last Glacial Maximum—as the authors state, the deep water bathing their

study site at the LGM changed from ~ 80% AABW and ~ 20% NADW to ~ 100% AABW (l. 150-151)—so assuming the modern vertical distribution for the Miocene is very sloppy and needs to be backed up by more than just “it works like that today”.

References

Samantha C. Carter, Elizabeth M. Griffith, Peter D. Clift, Howie D. Scher, Timothy M. Dellapenna; Clay-fraction strontium and neodymium isotopes in the Indus Fan: implications for sediment transport and provenance. *Geological Magazine* 2020;; 157 (6): 879–894.

Response To Reviewers' Comments

Manuscript No.: NCOMMS-21-41188A

Manuscript Title: Onset of modern-like deep water circulation in Indian Ocean caused by Central American Seaway closure.

REVIEWER COMMENTS

Reviewer #1 (Remarks to the Author):

The revised manuscript titled “Onset of modern-like deep water circulation in the Indian Ocean caused by Central American Seaway closure” by Prabhat et al. has addressed most of my concerns and questions. The authors have also improved the text as appeared from the track-changes document, which has significantly improved the principal tenor of the manuscript, which the authors have been struggling to convey.

Reply: We thank the reviewer for the comment.

One of the essential findings by Prabhat et al. is the demonstration of the integrity of the eNd values as a reliable deep-water circulation proxy proximal to the continental margin. The authors provided data on simultaneous extraction and measurements of eNd in fossil fish teeth, detrital and authigenic fractions, not an easy task or effort to overlook, primarily if terrestrial sediments were also deposited at the sediment core sites. For example, Naik et al. (2019 G3; their Figs. 2 and 4) demonstrated that the fossil fish teeth/broken bone fragments and foraminifers show insignificant differences in eNd values from a sediment core in the Bay of Bengal. Further, Naik et al. (2019) has demonstrated (their Fig. 4) insignificant eNd variability regardless of cleaning protocols such as cleaned foraminifers, decarbonated sediment leachates, acidic acid leachates, and uncleaned foraminifers being used. These earlier findings further support the validity of eNd extraction (Fig. S3) and hence, the eNd data of Lathika et al. (2021), one of the coauthors in Prabhat et al., as well as the current manuscript by Prabhat et al.

Reply: We thank the reviewer for the constructive comments. We also appreciate the reviewer for explaining the methods as an expert in this field and clarifying the specific points on the method's robustness.

I appreciate authors' elaboration of the discussion using *G. bulloides* (surface ocean proxy), but it reflects the strength of the summer monsoon and ϵNd (bottom water proxy, at the IODP site U1457) and how the onset of the circulation was coupled to the intensity of the Indian summer monsoon. This description was dormant, but the revised manuscript clarifies the linkage.

Reply: We thank the reviewer for the comment.

I am not fully onboard with their End-member calculation method of apportioning ϵNd values to the AABW and NADW as I raised it to their initial submission of the manuscript; however, my hesitancy should not prevent the manuscript from being published should Nat. Comm. decides to accept it.

Reply: We thank the reviewer for the constructive review.

I have edited the manuscript's abstract (see attached), but I suppose either Nat. Comm or someone else could improve the text further.

Reply: We thank the reviewer for suggestions and for improving the text. We have incorporated the changes in the "Abstract" of the revised version. We have thoroughly edited the text in the revised version to improve the readability.

Reviewer #2 (Remarks to the Author):

I appreciate the effort made by the authors to reply to each comment of the reviews. However, I regret to say that some responses to the major comments are in my opinion not satisfying and the new version of the manuscript pertaining to these comments is nowhere more convincing than the original.

Though I do not doubt the quality of the data presented in this new deep-sea record or reject the possibility that the authors' interpretation could be correct, I think the arguments presented in the manuscript are just too ambiguous or circular yet to firmly back up the authors' story. My recommendation is therefore to reject the manuscript in its current form.

1. I maintain my first major comment of the last round of reviews, given that the authors' answer simply repeats the argumentation presented in the manuscript and do not provide any other analysis that could help refute the alternative interpretation that I suggested. Again, I do not claim that I am correct but just that it is a possibility that should be dismissed and the authors have up to now failed to do so.

In particular, the authors answer states that "Fish teeth/debris ϵNd is considered to be more robust and reliable archive to reconstruct the water mass signature without detrital contamination (Frank, 2002; Martin and Haley, 2000)". But precisely, if fish teeth ϵNd is a reliable seawater signature not affected by detrital contamination, then how do the authors explain that fish teeth ϵNd is remarkably constant across the record? This instead suggests that the deep seawater ϵNd at this site remained remarkably similar between 11 and 2 Ma—note that this does not mean that deep sea currents did not change, only that any "new" current carried almost the same ϵNd signature. To be fair, I follow the authors that detrital ϵNd was affected by competition between dust inputs (more radiogenic) and riverine inputs (less radiogenic) from westward flowing Himalayan rivers, the ϵNd signature of which decrease between 10 – 2 Ma (Carter et al. 2020). I do not contest their interpretation of increased aridification at 7.4 – 6 Ma, ~ 3.5 Ma and 2.5 – 2 Ma, leading to dust-dominated intervals in the detrital record that exhibit radiogenic excursions (as the authors show, these dust-dominated periods are coeval with low sedimentation rate suggesting a decreased riverine flux and thus a more radiogenic ϵNd signature in the detrital record). My view diverges in that I suggest that at times of low sedimentation rate (i.e. the dust-dominated intervals excluded by the authors), which are incidentally also times of higher productivity (most likely marine but this is unclear), the proportion of detrital material incorporated in the bulk sediment decreases. Therefore, the authigenic ϵNd signature obtained from the bulk sediment is less likely to be affected by partial dissolution of detrital material during the leaching, and thus the authigenic ϵNd signature gets closer to the seawater signature. This is consistent with the fact that authigenic ϵNd and fish teeth ϵNd are similar during these intervals. Conversely, at times of high sedimentation rate (for which there are unfortunately no fish teeth ϵNd data), the detrital material makes up for a much larger proportion of the bulk sediment and the authigenic ϵNd is more likely affected by partial dissolution of detrital material. This acts to draw the authigenic ϵNd

signature closer to the detrital signature and explains the decrease seen in the authigenic ϵ_{Nd} record.

Reply: In view of the comments on the possible detrital contamination, we have added discussion in the revised manuscript (Line no. 126 to 151) and ruled out the possibility of partial dissolution of detrital material during the leaching.

Considering the above comments, we have provided a schematic in the supplementary (Figure S5) to explain the authigenic ϵ_{Nd} signal record by the fish teeth debris and sediment leached. Please see the same figure below. Fish teeth always records the deep-water signature after the deposition. In the normal scenario, the deep-seawater ϵ_{Nd} signature reflects the circulation and mixing signature of AABW and NADW (Figure S5a). However, in extreme cases in two different scenarios, the deep-water ϵ_{Nd} signature could be contaminated by the dissolution of sediments and will be dragged towards the detrital ϵ_{Nd} signature i.e.

(i) First scenario relates to the release of particulate Nd supplied by the supplied river during the high monsoonal discharge (Figure S5b). This scenario has been demonstrated by Naik, et al. ¹ for the Bay of Bengal records; deep water ϵ_{Nd} signature was dragged towards the particulate ϵ_{Nd} value due to the release of particulate Nd. However, radiogenic excursion due to such processes in the Arabian Sea is unlikely because radiogenic sediment sources are only derived from the river Tapi², which has much lower sediment discharge compared to other major west flowing rivers (e.g. Narmada and Indus) supplying less radiogenic sediments. The study of modern water column³ and past deep water circulation of the last glacial cycle (~136 ka)⁴ have demonstrated that such processes are insignificant and therefore, the variability in the ϵ_{Nd} can only be explained in terms of deep water circulation.

(ii) The second possibility is the high dust deposition over the Arabian Sea (Figure S5c); dust can be dissolved in the water column during settling through the water column. In the normal scenario, dissolution of dust and its contribution is restricted near the surface (~100 m), as demonstrated by Goswami, et al. ³. However, during episodic high dust events, dust dissolution can impact deep water, which can drag deep water ϵ_{Nd} signature towards the Arabian Sea dust characterised by radiogenic Nd (ϵ_{Nd} ~-5). The excursions with similar radiogenic ϵ_{Nd} values during the arid

intervals of 7.4 – 6 Ma, ~ 3.5 Ma and 2.5 – 2 Ma indicate high dust deposition with constant dust source. Therefore, fish teeth ϵ_{Nd} will reflect the deepwater signature contaminated by dust dissolution.

Figure. An illustration of how deep water acquires ϵ_{Nd} values in different scenarios and archived its signature in the authigenic phase. For details, please see the supplementary Figure S5 and its captions.

In summary, the authors would like to exclude the dust-dominated intervals at 7.4 – 6 Ma, ~ 3.5 Ma and 2.5 – 2 Ma because of potential terrestrial contamination, whereas I suggest that these intervals instead better record the deep sea ϵ_{Nd} signature.

Reply: Thanks for the comments and for providing the alternate thoughts for explaining these radiogenic excursions in terms of deep water circulation with a new current/water mass with similar ϵ_{Nd} values. We have explored such possibility. The only possible radiogenic water mass (with ϵ_{Nd} -4) is the water masses from the Pacific Ocean, which has been discussed in detail in the manuscript (Line no 274 to 289). However, our investigation did not find any evidence to find such possibility (Line no. 199 to 216 and 348 to 365).

2. In the response to my other major points, I have a problem with the fact that the authors invoke the modern ocean deep water configuration to justify what they observe (or calculate in the case of the binary mixing model) in the late Miocene.

The issue is that assuming that modern vertical water mass distribution is the same in the Miocene is somewhat a circular reasoning if the aim is to document deep water evolution towards modern conditions. The vertical distribution of water mass is not even known for sure at the Last Glacial Maximum—as the authors state, the deep water bathing their study site at the LGM changed from ~ 80% AABW and ~ 20% NADW to ~ 100% AABW (l. 150-151)—so assuming the modern vertical distribution for the Miocene is very sloppy and needs to be backed up by more than just “it works like that today”.

Reply: Thank you for the comment. We have added discussion in the revised manuscript (Line no. 330 to 347) and a figure in the supplementary (Figure S8). Our proxy based reconstruction and its interpretation is restricted to transport pathways and volume transport. A recent study⁵ based on the compilation of calcium carbonate records from various depths in the entire Atlantic basin provides an idea about the past lysocline depth and calcium carbonate compensation depth (CCD) which reflect changes in the vertical distribution of deep-water masses. The ϵ_{Nd} based reconstruction of volume transport is consistent with the changes in lysocline and CCD in the Atlantic and thus corroborates our conclusion.

Figure. Showing the increased carbonate preservation in South Atlantic and throughout the Atlantic Ocean with enhanced NCW transport and the deepening of the carbonate compensation depth caused by enhanced NCW since late Miocene⁵. For details, please see the supplementary Figure S9.

References:

- 1 Naik, S. S., Basak, C., Goldstein, S. L., Naidu, P. D. & Naik, S. N. A 16-kyr Record of Ocean Circulation and Monsoon Intensification From the Central Bay of Bengal. *Geochemistry, Geophysics, Geosystems* **20**, 872-882, doi:10.1029/2018gc007860 (2019).
- 2 Goswami, V., Singh, S. K., Bhushan, R. & Rai, V. K. Temporal variations in $^{87}\text{Sr}/^{86}\text{Sr}$ and ϵNd in sediments of the southeastern Arabian Sea: Impact of monsoon and surface water circulation. *Geochemistry, Geophysics, Geosystems* **13**, n/a-n/a, doi:10.1029/2011gc003802 (2012).
- 3 Goswami, V., Singh, S. K. & Bhushan, R. Impact of water mass mixing and dust deposition on Nd concentration and ϵNd of the Arabian Sea water column. *Geochimica et Cosmochimica Acta* **145**, 30-49, doi:10.1016/j.gca.2014.09.006 (2014).
- 4 Lathika, N. *et al.* Deep water circulation in the Arabian Sea during the last glacial cycle: Implications for paleo-redox condition, carbon sink and atmospheric CO₂ variability. *Quaternary Science Reviews* **257**, 106853, doi:<https://doi.org/10.1016/j.quascirev.2021.106853> (2021).
- 5 Keating-Bitonti, C. R. & Peters, S. E. Influence of increasing carbonate saturation in Atlantic bottom water during the late Miocene. *Palaeogeography, Palaeoclimatology, Palaeoecology* **518**, 134-142, doi:<https://doi.org/10.1016/j.palaeo.2019.01.006> (2019).